# EgoPlan: Towards Effective Embodied Agents via Egocentric Planning

## Abstract

We explore leveraging large multi-modal models (LMMs) and Text2image models to build a more general embodied agent. LMMs excel in planning long-horizon tasks over symbolic abstractions but struggle with grounding in the physical world, often failing to accurately identify object positions in images. A bridge is needed to connect LMMs to the physical world. The paper proposes a novel approach, egocentric vision language planning (EgoPlan), to handle long-horizon tasks from an egocentric perspective in varying household scenarios. This pipeline leverages a diffusion model to simulate the fundamental dynamics between states and actions, discusses how to integrate computer vision related techniques like style transfer and optical flow to enhance ability of modeling spatial states and generalization across different environmental dynamics. The LMM serves as a planner, breaking down instructions into sub-goals and selecting actions based on their alignment with these sub-goals, thus enabling more generalized and effective decision-making. By using LMM, we can output text actions, using a series of mechanisms such as reflection to perform high-level task decomposition and low-level action output end-to-end. Experiments show that EgoPlan improves long-horizon task success rates from the egocentric view compared to baselines across household scenarios.

## 1 Introduction

The advent of large language models (LLMs) (et al., 2024b; Touvron et al., 2023) and large multi-modal models (LMMs) (202, 2023; Girdhar et al., 2023; Zhang et al., 2023a; Zhu et al., 2023) has revolutionized the field of artificial intelligence. Their strong reasoning (Wang et al., 2023b; Wei et al., 2023) and powerful generalization capabilities allow them to be directly applied in various scenarios. In the next step toward artificial general intelligence (AGI), researchers are considering enabling large models (LMs), especially LMMs, to break through the world expressed by text and images to interact with the physical world. They aim to build a general embodied agent that intelligently interacts with the physical world.

LMMs exhibit impressive long-horizon planning over symbolic abstractions (Wake et al., 2024), yet struggle with grounding text in the physical world, often failing in precise object localization. While LMMs understand what to do, they lack understanding of how the world functions, necessitating a world model to bridge this gap. Two potential solutions exist: implicitly integrating dynamics via extensive fine-tuning on state-action sequences (Driess et al., 2023; et al., 2023), which demands substantial resources, or explicitly employing pre-trained generative world models (e.g., Text2image/video) as auxiliary tools (Radford et al., 2021; Saharia et al., 2022). Prior work (Black et al., 2023; Du et al., 2023b) suggests that these models can inject world knowledge by predicting future observations or trajectories. This work investigates the latter approach, exploring the potential of leveraging LMMs and Text2image models for more general embodied agents.

Existing approaches (Du et al., 2023a; Zhou et al., 2024) using Text2image/video models as world models for decision-making face limitations. First, their focus on fully observable object manipulation tasks is atypical of real-world scenarios and their adaptability to partially observable settings is unclear. For instance, methods requiring multi-step image generation (Black et al., 2023; Du et al., 2023b) suffer from error accumulation in partially observed environments like autonomous driving. Second, their frameworks exhibit limited capability in: (i) task-specific low-level policies with potential for collapse upon new dynamics; (ii) coarse-grained text action representations hindering the mapping

to fine-grained state transitions, especially in complex, partially observable tasks; and (iii) the lack of individual entity motion patterns, limiting generalization to novel environments with different dynamics within the same task category. We aim for generalization to varying dynamics within fixed household scenarios.

To address these limitations, we propose Egocentric Vision Language Planning (**EgoPlan**), a general embodied agent for long-horizon egocentric tasks. Recognizing the rich action and state transition information in optical flow (Ko et al., 2023; Yang & Ramanan, 2020), we integrate it into our world model for enhanced spatial understanding in navigation and object motion prediction in manipulation, contrasting with traditional text-based actions. Furthermore, to handle visual style variations across different simulated home environments, we employ LoRA fine-tuning to enable adaptation to diverse visual distributions while preserving learned motion patterns. This enhances fine-grained texture modeling and generation across scenes, allowing for transfer to new environments with limited samples, achieving a style transfer-like effect.

We conduct a comprehensive evaluation and analysis of each module of the embodied agent. Empirically, we demonstrate the high quality of image generation by the dynamics model and the high accuracy of optical flow prediction. Subsequently, we verify the dynamics model's effectiveness in aiding decision-making in more complex tasks. Lastly, we confirm the method's generalization capabilities in a different environment. Our major contributions are summarized as follows:

- We have collected a dataset on Virtualhome, which views high-level manipulation/navigation actions of the agent in Virtualhome as trajectorys and provides egocentric observations each time-step and fine-grained action information, which will provide data support for navigation and manipulation tasks in the embodied environment. See Section 3 for details.

- We propose **EgoPlan**, a framework for complex task planning that combines LMM and a dynamics model that predicts an egocentric view of the next time step and the subgoal is completed. We also introduce optical flow into the dynamics model and borrow the idea of style transfer in computer vision and adopt the LoRA (Hu et al., 2021) model to achieve few-shot generalization in different embodied scenarios.

- For the action selection and decision-making module, we employ the LMM as the execution module in both the high-level task decomposition and low-level action selection components. We utilize a series of reflection and summarization mechanisms to accomplish tasks, while also ensuring the agent inherits this ability of generalizing the downstream polices to new dynamics. Experiments on comprehensive tasks demonstrate the effectiveness of our framework through LMM+dynamics model planning.

## 2 RELATED WORK

In this section, we present a brief overview of related work. More discussions are in Appendix B.

### 2.1 DYNAMIC MODEL AND WORLD MODEL FOR DECISION-MAKING

The world model is used to model the dynamics of the environment. It is crucial for building autonomous agents and enabling intelligent interactions in various scenarios. However, developing a precise world model remains a significant challenge in model-based decision-making. The advancements in diffusion-based world models are reshaping how we model physical motion laws in real-world settings, particularly in robotics. UniPi (Du et al., 2023a) frames the decision-making problem in robotics as a Text2video task. The generated video is fed into an inverse dynamics model (IDM) that extracts underlying low-level control actions, which are executed in simulation or by a real robot agent. Video Language Planning (VLP) (Du et al., 2023b) introduces a novel method for task planning that integrates video generation with tree search algorithms. This methodology lets robots plan over longer horizons by visualizing future actions and outcomes. Unlike previous works, SuSIE (Black et al., 2023) leverages pre-trained image-editing models to predict the hypothetical future frame. A low-level goal-reaching policy is trained on robot data to reach this hypothetical future frame. Since one goal frame prediction does not require the model to understand the intricacies of the robot's low-level precisely dynamics, it should facilitate transfer from other data sources such as human videos. RoboDreamer (Zhou et al., 2024) advances the field by utilizing video

(a) observation    (b) next observation    (c) seg_inst    (d) depth    (e) optical flow

Figure 1: An illustration sample in VH-1.5M, which includes current image observation, next image observation given the text action, semantic segmentation map, depth map, and optical flow map.

diffusion to formulate plans combining actions and objects, solving novel tasks in unexplored robotic environments.

## 2.2 EMBODIED AGENT WITH LMMS

Recent methods use LMMs to assist planning and reasoning in simulation environments (Fan et al., 2022; Wang et al., 2023a; Yao et al., 2023)and robot learning (Ahn et al., 2022; Liang et al., 2023; Zeng et al., 2022). LMMs are also applied to help robot navigation (Parisi et al., 2022; Majumdar et al., 2020) and manipulation (Jiang et al., 2022; Ren et al., 2023; Khandelwal et al., 2022). Among them, ReAct (Yao et al., 2023) uses chain-of-thought prompting by generating both reasoning traces and action plans with LMMs. SayCan (Ahn et al., 2022) leverages the ability of LLMs to understand human instructions to make plans for completing tasks without finetuning LLMs. Voyager (Wang et al., 2023a) leverages GPT-4 to learn and continually discover skills during learning. While these studies demonstrate encouraging outcomes, they depend significantly on the inherent capabilities of powerful large language models (LLMs), which poses challenges for their application to smaller language and multimodal models (LMMs) with limited reasoning abilities.

## 3 VH-1.5M DATASET

Existing vision-language-action datasets, such as RT-X (et al., 2024a) and RH20T (Fang et al., 2023), often utilize static views to mitigate perspective change issues, providing "fixed camera" observations suitable for fully observable task planning where all manipulable objects are within a constant field of view. In contrast, datasets like ALFRED (Shridhar et al., 2020) and ProcTHOR (Deitke et al., 2022) employ egocentric views, introducing significant perspective changes and necessitating embodied task planning under partial observability, where current observations may be insufficient for task completion without viewpoint adjustments for information gathering (e.g., navigating to an unseen object). Our dataset distinguishes itself from other embodied navigation datasets by incorporating agent motion trajectories (e.g., grasp, put) and the coupled perspective and hand position changes. To address the aforementioned limitations, we introduce the VH-1.5M dataset, built upon the VirtualHome environment (Puig et al., 2018; 2020).

We construct our dataset VH-1.5M in the VirtualHome environment, which comprises 50 distinct houses. Each house contains approximately 300 interactive objects, and the embodied agent can perform more than 10 actions. Note that the VirtualHome environment is a simulator tailored for embodied agents, offering a detailed simulation of a residential living scenario. It enables a range of household tasks, such as navigation and object manipulation.

The VH-1.5M dataset is organized in a structured manner, encapsulating the relationship between actions, houses, agents, and trajectories. Each task sequence entry follows a hierarchical structure, for example: "/open/house_0/Female4/2_fridge" (female4 open the fridge2 in house0).

**Dataset Details:** The VH-1.5M dataset consists of:

- 13 Actions: Various physical actions and interactions for agents within the houses. These action instructions are high level and can be completed in a sequence of time steps, such as walk to microwave.
- 50 Houses: Uniquely designed houses with diverse layouts and object placements.
- 4 Agents: Four distinct agents (simulated humans), each capable of performing the full range of actions.
- 1.5M Samples: Dateset has numerous detailed sequences, each executing one action instruction. Information from each step in the sequence is stored as one sample. One example is shown in Figure 1.

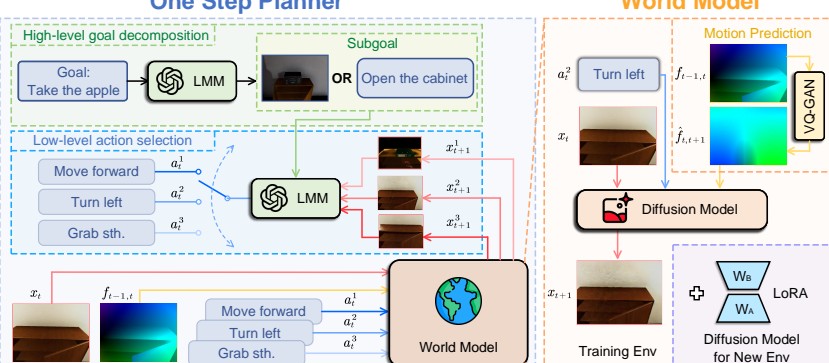

Figure 2: Overview of EgoPlan. The left side features a one-step planner that provides the agent with decision-making capabilities, while the right side includes a world model (dynamics model) that provides the agent with an understanding of the current environment.
More details of the dataset can be found in the Appendix D, and **we will open-source the dataset.**

# 4 METHOD

Our embodied agent, EgoPlan, takes visual observation $x_t$ of the scene at the current timestep $t$ and a natural language goal $g$ as inputs and outputs an action $a_t$ to interact with the environment. Note that the $x_t$ only partially represents the current environment state. In addition, the agent uses encapsulated skills as actions, such as moving forward, turning, and grabbing objects. For problem settings, the decision-making environment is typically characterized as a Partially Observable Markov Decision Process (POMDP) (Smallwood & Sondik, 1973), defined as a tuple $(\mathcal{O}, \mathcal{A}, p, r, \gamma)$. In our pipeline, we define egocentric observation $x_t$ as partial observation $\mathcal{O}$, and we will train a text2image model as dynamic model to model dynamic processes $p(o_t|o_{t-1}, a_{t-1})$. For Egoplan agent, we model the actions in a Markov process using either textual descriptions as $a_t = l_t$ or a more fine-grained description of actions: optical flow, which can be denoted as $a_t = f_{t,t+1}$.

EgoPlan consists of two parts, as illustrated in Figure 2. The first is a dynamics model that gives the agent the concept of the current environment, and the other is the planner that endows the agent with decision-making capabilities. Intuitively, we humans first envision the outcomes of each action in our minds, and then, by comparing the results, we make the best decision. In the same way, we use a dynamic model to create an egocentric scenario where different actions can be taken, which is then fed into LMM to determine which action is more reasonable.

## 4.1 DIFFUSION-BASED DYNAMICS MODEL

### 4.1.1 LEARNING DYNAMICS

From a first-person perspective, the view after two or more steps may be completely different, making it difficult to model. Therefore, we aim to model the fundamental dynamics model, $p_\theta(x_{t+1}|x_t, a_t)$, for one-step planning usage. In more detail, we want to generate a new image $x_{t+1}$, representing the next state given the current visual observation $x_t$ and the text of the action $a_t$. Then, we cast our eyes on the Text2image model and resort to the diffusion model for modeling specifically. It has an irreplaceable advantage in easily incorporating other modalities as a condition.

Although the open-sourced diffusion model (Ho et al., 2022; Luo et al., 2023), $p_\theta(x_{tar}|x_{src}, l)$, trained on a wealth of online videos, has demonstrated the ability to predict the future, their generated results are hard to control, and most are only semantically reasonable. Moreover, most of the text in the pre-trained dataset consists of image descriptions $l$ rather than action instructions $a$. Therefore, supervised fine-tuning is adopted based on our VH-1.5M dataset to better model the dynamics, $p_{\theta_{sft}}(x_{t+1}|x_t, a_t)$. Formally, the training objective is given by:

$$\mathcal{L}_{\text{MSE}} = \left\| \epsilon - \epsilon_\theta \left( q\left( x_{t+1}^{(k)} | x_t, a_t \right), k \right) \right\|^2 = \left\| \epsilon - \epsilon_\theta \left( \sqrt{\overline{\alpha_t}} x_t + \sqrt{1 - \overline{\alpha_t}} \epsilon | a_t \right) \right\|^2 \quad (1)$$

where $\epsilon_\theta$ is a learnable denoising model for reverse process, $k$ is denoising steps, and $\overline{\alpha_t}$ are a set of $K$ different noise levels for each $k \in [1, K]$, and $x_t, a_t$ separately represent the current observation image and action description text. In practice, we'll use Instructpix2pix as the backbone network; see the Appendix I for training details. However, we find it difficult to generalize directly to other

environments since our dataset only includes VirtualHome scenes. The difference between two environments, such as Habitat 2.0 (Savva et al., 2019; Szot et al., 2022) and VirtualHome, primarily lies in their different motion patterns for the same action and distinct visual styles. Especially for the former, the motion pattern, such as the amplitude of the same action, performed by agents in a different environment can be unpredictable.

### 4.1.2 GENERALIZATION

We want to improve the model's generalization ability from a different perspective. In other words, instead of enhancing generalization through big data and large models, we aim to explicitly address the differences between environments such as the visual style of indoor environments and the definition of action amplitudes at the methodological level.

**Motion Regularization.** Firstly, we must combine the motion information into the diffusion model to distinguish the different motion patterns. Optical flow has thus caught our attention. It refers to the pattern of apparent motion of image objects between two consecutive frames caused by objects or camera movement. In optical flow maps, colors represent the direction of motion, and the depth or intensity of the colors indicates the magnitude of the motion, which is a general feature across different environments.

However, in practice, in the absence of the next observation, we cannot obtain the current optical flow, $f_{t,t+1}$. Inspired by other motion estimation works (Chen & Koltun, 2016; Zach et al., 2007), we assume motion consistency holds over short intervals, meaning abrupt changes do not occur. Consequently, the consecutive optical flow maps are highly correlated, allowing us to predict the current optical flow map using the previous map. The previous map is calculated from the previous two frames and reflects the actual motion pattern in the current environment.

We notice that optical flow generation does not require complex texture generation, and it is expected not to cause a significant delay in the pipeline. Therefore, we adopt a less powerful but lightweight generative model, VQ-GAN (Esser et al., 2021), and train it on our dataset to predict the optical flow map. Empirically, the generalization ability to predict optical flow is much better than predicting actual images. Formally, the training objective is given by:

$$\min \mathcal{L}_{VQ}(E, G, Z) = \|x - \hat{x}\|_2^2 + \|\text{sg}[E(x)] - z_q\|_2^2 + \beta\|\text{sg}[z_q] - E(x)\|_2^2,$$

where $E$ is the encoder, $G$ is the generator, $Z$ represents the latent space, $x$ is the input image, $\hat{x}$ is the reconstructed image, $z_q$ is the quantized latent vector, sg denotes the stop-gradient operator, and $\beta$ is a hyperparameter that balances the commitment loss.

*In summary, we use a simple model to predict motion patterns and then a more complex model to reconstruct real textures based on motion patterns.* Therefore, we adopt ControlNet (Zhang et al., 2023b) to incorporate the optical flow map, $f_{t,t+1}$, into the default diffusion model, $p_{\theta_{\text{sft}}}(x_{t+1}|x_t, a_t, f_{t,t+1})$. Only the ControlNet part needs to be trained on VH-1.5M at this stage. The training details of VQ-GAN and ControlNet can be found in Appendix I. Formally, the training objective is given by:

$$\mathcal{L}_{\text{MSE}} = \left\| \epsilon - \epsilon_\theta \left( q\left( x_{t+1}^{(k)}|x_t, a_t, f_{t,t+1} \right), k \right) \right\|^2 \tag{2}$$

$$= \left\| \epsilon - \epsilon_\theta \left( \sqrt{\overline{\alpha_t}}x_t + \sqrt{1 - \overline{\alpha_t}}\epsilon | a_t, f_{t,t+1} \right) \right\|^2. \tag{3}$$

**Style Transfer.** Secondly, we use LoRA to fine-tune the diffusion model for visual style transfer. Note that LoRA requires very little data, just about 20 of samples. Normally, it is convenient to collect data on such a scale in new environments. We expect the model to achieve generalization with as little effort as possible. In Section 5.2, we can find the role of LoRA method in maintaining the action pattern of the model between different environments, while flexibly transferring the style of fine-grained observation images.

### 4.2 PLANNING WITH DYNAMICS MODEL

To avoid further training in new environments, we prompt the LMM GPT-4V, as the planner. The LMM needs to be responsible for high-level goal decomposition as well as low-level action selection. Meanwhile, the pre-trained dynamics model can help LMM better understand the world.

|  (a) Original | (b) InstructP2P (finetuned) | (c) Previous flow | (d) Ours (previous flow) | (e) Predicted flow | (f) Ours |

Figure 3: Examples of the generated image of the next observation in VirtualHome. The tasks from rows 1 to 4 are: close the fridge, switch off the light, turn left, and turn right.

### 4.2.1 GOAL DECOMPOSITION

For long-term complex tasks, subgoal decomposition is crucial. Subgoals can be represented as text ($g_{\text{tar}}$) or images ($x_{\text{tar}}$). For text-based subgoals, we prompt the LMM for a plausible one. Additionally, we train a diffusion model, $p_{\theta_{\text{sft}}}(x_{\text{tar}}|x_t, g_{\text{tar}})$, to generate image-based subgoals conditioned on the text subgoal and current observation. While prior work (Black et al., 2023; Zhou et al., 2024) uses diffusion models serially to predict subgoal state images for long-horizon manipulation tasks, generating subgoal scene images for composite manipulation and navigation tasks, particularly navigation, presents a greater challenge. This is due to the substantial changes in the entire image scene and the joint positions of numerous objects required, demanding a strong understanding of spatial attributes beyond simple object-centric image editing. Consequently, predicting subgoal images can be less precise than predicting the next observation. We plan to investigate the impact of different subgoal types on task performance (Section 5.4).

### 4.2.2 ONE-STEP PLANNER

Since we can only ensure that the prediction for the next step is relatively accurate, we adopt a one-step planning method. In more detail, we utilize the pre-trained dynamics model to predict the visual outcomes of all the actions in the next state. Once the text/image-based subgoal is obtained, we send the subgoal and all the visual outcomes to the LMM. Then, we prompt it to compare all the potential outcomes with the subgoal and determine which action can bring the agent closer to the goal. So the process of goal decomposition and one-step planner is equivalent to the following formula.

$$\{G_0, G_1, \cdots, G_n\} = \text{LMM}(s_0, task) \tag{4}$$

$$a^* = \arg\min_{a \in A} d\left(f(s_t, a), G \in \{G_0, G_1, \cdots, G_n\}\right) \tag{5}$$

In the aforementioned equations, $\{G_0, G_1, \cdots, G_n\}$ refers to a series of subgoals that are decomposed from the task using LMM. $f$ is the dynamic model and $d$ is the distance metric function, in our pipeline, GPT4V judges how far the target is from the dynamic model prediction. It is noteworthy that, in selecting the optimal action for one-step planning process, inspired by Tan et al. (2024); Zhai et al. (2024), we utilize LMM to generate low-level actions in contrast to reinforcement learning or imitation learning algorithms. In this context, we leverage the comprehension capabilities of LMM to ensure the generalization of the low-level action in cross-environment decision-making. We also employing mechanisms like React (Yao et al., 2023) and Reflexion (Shinn et al., 2023) to enhance the agent's performance, which are shown in Appendix H. The prompt of task-decomposition and low-level action selection has been listed in Appendix G. Black et al. (2023) has discussed the generalization of objects concerning various operational targets; however, the generalization of underlying policy networks based on reinforcement learning or imitation learning algorithms, particularly in response to changes in the entire environmental scene—especially in navigation tasks, the ability of the pipeline still requires improvement. We will further discuss the experimental outcomes related to this in Sections 5.2 and 5.4.

(a) Enclose the fridge  (b) Go through door  (c) Shut off the PC  (d) Take hold of pillow (e) Switch off the light  (f) Shut the stove  (g) Open the cabinet

Figure 4: Examples of the generated image subgoals. The first row is the original image, and the second row is the image subgoal generated based on the text subgoal.

## 5 EXPERIMENT

In this section, we comprehensively evaluate and analyze each module of the embodied agent. We first evaluate the quality of image generation using the world model and the quality of optical flow prediction. Secondly, we evaluate whether our world model can assist task planners in completing more complex tasks. Finally, we assess the generalization of our method. In addition to the below experiments, we also do a series of works to discuss the **complexity of the system** to explain why we do one-step planning. See the Appendix J for detailed analysis.

### 5.1 VISUAL QUALITY

We adopt two metrics, FID (Heusel et al., 2018) and user score, to evaluate the visual quality of the generated image of the world model. For models, **InstructP2P (pre-trained)** is the default model of InstructP2P. **InstructP2P (fine-tuned)** is the model fine-tuned on our dataset. **Ours (previous flow)** is the world model that conditions on the previous optical flow map, while **Ours** is conditioned on the predicted optical flow map. Note that the validation set of VH-1.5M has around 5k samples.

Table 1: FID score comparison with other models on the validation set. It is calculated between the predicted observation and ground truth. The lower the number, the better the quality of the image.

| Model | Mean | Variance |
|---|---|---|
| InstructP2P (pre-trained) | 13.65 | 0.10 |
| InstructP2P (fine-tuned) | 1.06 | 0.05 |
| Ours (previous flow) | 0.83 | 0.03 |
| Ours | **0.82** | 0.03 |

**FID Score.** FID is a standard metric measuring the distance of two image distributions using the inception model. The smaller the FID is, the more similar the two images are. Table 1 shows the FID score of our model and baselines. We can see that using existing diffusion models as world models is ineffective because their training data often lacks state transition-related data. Meanwhile, introducing an optical flow map, which serves as motion pattern information, significantly enhances the generation results. In addition, world models based on predicted optical flow are slightly better than those based on the optical flow of the previous frame.

**User Study.** We also conduct a user study on the accuracy of world models for image generation. For the criterion, users judge the correctness of the direction and amplitude of the executed action. Each user investigates a total of 1000 samples from the validation set. There are 8 users participating in the survey in total. Our user study, shown in Table 2, again verifies our predicted optical flow can help generate higher-quality images.

Table 2: User score of the user study. The user score is the percentage of images that users consider to meet the criteria out of the total 1000 images. The higher the number, the better the quality of the image. The evaluated images are from the validation set.

| Model | Mean | Variance |
|---|---|---|
| InstructP2P (fine-tuned) | 54.10% | 1.53% |
| Ours (previous flow) | 69.35% | 1.34% |
| Ours | **74.93%** | 2.57% |

**Analysis.** As illustrated in Figure 3, InstructP2P (fine-tuned) generates the scene of steering in the wrong direction. However, this flaw can be greatly improved by incorporating optical flow information. Moreover, it is observed that the dynamics of closing the refrigerator can be more accurately predicted if the prediction of the motion pattern is considered. More examples can be seen in Appendix E.

### 5.2 VIRTUALHOME TASKS

**Results.** To demonstrate that our world model can well assist the LMM in task planning, we evaluate various methods on 12 tasks in VirtualHome environment, each task described by an instruction and can be broken down into a number of subtasks. Each task instruction, subtasks and some experiment

Table 3: The average length of completed subtasks on 12 tasks for all the methods. Tasks 1-6 occur inside one room, while tasks 7-12 take place in two rooms. This metric measures the average number of subtasks completed per execution after 100 executions of each task. **We reported the task completion rate in the Appendix K.**

| | GPT4+React | GPT4V | React | Reflexion | GPT4V+P2P | GPT4V+OF | SuSIE | GR-MG | Ours(text goal) | Ours(image goal) |
|---|---|---|---|---|---|---|---|---|---|---|
| take and place | 0.11 | 0.26 | 0.57 | 0.87 | 1.21 | 1.64 | 1.42 | 1.61 | 1.63 | **1.88** |
| take and put1 | 0.12 | 0.34 | 0.65 | 0.80 | 1.22 | 1.34 | 0.98 | 1.68 | 1.75 | **2.02** |
| take and put2 | 0.21 | 0.34 | 0.59 | 0.76 | 1.32 | 1.47 | 1.41 | 1.63 | 1.69 | **1.91** |
| take and drink | 0.08 | 0.46 | 0.81 | 0.79 | 1.19 | 1.47 | 1.39 | 1.77 | 1.99 | **2.11** |
| turn on sit | 0.10 | 0.31 | 0.75 | 0.81 | 1.29 | 1.51 | 1.31 | 1.68 | 1.71 | **2.00** |
| put apple | 0.09 | 0.35 | 0.69 | 0.97 | 1.18 | 1.61 | 1.69 | 1.86 | **1.93** | 1.97 |
| take and place2 | 0.16 | 0.45 | 0.66 | 0.96 | 1.28 | 1.50 | 1.23 | 1.75 | 1.81 | **1.88** |
| take and place3 | 0.17 | 0.34 | 0.63 | 0.86 | 1.14 | 1.57 | 1. | 1.61 | 2.05 | **2.12** |
| take and put3 | 0.15 | 0.33 | 0.74 | 0.96 | 1.11 | 1.55 | 1.17 | 1.83 | 1.81 | **2.01** |
| take open and put | 0.12 | 0.38 | 0.64 | 0.84 | 1.22 | 1.46 | 1.15 | 1.93 | 1.77 | **1.99** |
| take put and open | 0.12 | 0.29 | 0.66 | 0.87 | 1.30 | 1.58 | 1.62 | 1.74 | 1.89 | **1.96** |
| take and put4 | 0.21 | 0.35 | 0.71 | 0.86 | 1.28 | 1.68 | 1.56 | 1.64 | 1.69 | **1.81** |

details can be found in Appendix C. Each task is tested 100 times, and the maximum step in one episode is 80. For each of the 12 tasks, we abbreviated the task names for convenience. For example, the instruction of task 1, "take the bread from the toaster and place it on the plate on the table," consists of four subtasks: a) walk to the toaster, b) grab the bread, c) walk to the plate, and d) place the bread on the plate. We use "take and place" to refer to task 1.

These 12 instructional tasks are comprised of multiple sequential sub-tasks. For baselines, we use GPT4 combined with React (Yao et al., 2023) as the task planner and policy, denoted as **GPT4+React**, and it takes input as the JSON format text environment description. We also directly use GPT-4V to make decisions, denoted as **GPT4V**, and we also combined GPT4V with React (Yao et al., 2023) and Reflexion (Shinn et al., 2023) as the task planner and policy. When employing the Reflexion algorithm, its actor component is based on the React algorithm. These two baselines are denoted as **React** and **Reflexion**. For ablation baselines, we use the fine-tuned InstrctP2P as the world model, denoted as **GPT4V+P2P**. The world model that conditions on the previous optical flow map is denoted as **GPT4V+OF**.

As shown in Table 3, the dynamic model significantly improves the GPT-4V ability on various long-horizon tasks. Moreover, the inclusion of optical flow information enhances the accuracy of image generation and further improves task planning performance. The results also demonstrate the effectiveness of the predicted optical flow map.

**Image Subgoal *vs*. Text Subgoal.** In this part, we analyze the impact of different types of subgoals on tasks. During the goal decomposition process, the text subgoal directly outputted by the LLM task planner represents a high-level, coarse-grained description. If our method can generate images of the scene at the completion time of the subgoal, a more detailed, fine-grained description can be obtained. This might enhance the action selection ability that relies on the quality of the subgoal.

When employing images as subgoals, our approach contrasts with methods like SuSIE (Black et al., 2023) and GR-MG (Li et al., 2025), which generate actions based on these subgoals using a one-step planning world model. Instead, we leverage an LMM for an end-to-end pipeline encompassing both task decomposition and action selection, diverging from SuSIE's goal-conditioned behavioral cloning (GCBC) for low-level policy and GR-MG's goal-conditioned vision-language-action model. As shown in Table 3, our method (denoted as **Ours**) outperforms SuSIE (**SuSIE**) and GR-MG (**GR-MG**), particularly in long-horizon composite task planning involving substantial perspective shifts and the necessity for subgoal reasoning. The robustness and efficacy of our pipeline for extended tasks are further evaluated through a comparison of completion rates against several baselines on VirtualHome tasks, detailed in Appendix K.

Specifically, we have trained an InstructP2P model based on VH-1.5M to generate the image when the subgoal is completed, with the generation results illustrated in Figure 4. The decision-making results in Table 3 show that fine-grained subgoal description is better than coarse-grained description, even if the generated image is not that accurate.

We also conduct a user study to evaluate the visual quality of the generated image-based subgoals. More details can be found in the Appendix F.

**Real-world Experiments.** We also conduct real-world experiments. We use the **Qwen2.5-VL** models (Bai et al., 2025) and GPT4V as the LMMs for the experiments, and compared the results of **Cosmos** (Agarwal et al., 2025) as the world model. At the same time, we also compared the success rates of a series of reinforcement learning and imitation learning methods in terms of tasks. Detailed in Appendix L.

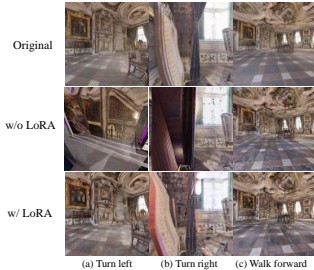

Figure 5: Examples of the generated images of the next observation in Habitat 2.0.

Table 4: We report the zero-shot evaluation results on the HM3D ObjectNav task. Comparison with state-of-the-art methods on the ObjectNav task.

| Method | with Mapping | LLM | Extra Sensors | SR | SPL |
|---|---|---|---|---|---|
| L3MVN (Yu et al., 2023) | with | GPT-2 | Depth, GPS | 35.2 | 16.5 |
| PixelNav (Cai et al., 2023) | without | GPT-4 | - | 37.9 | 20.5 |
| ESC (Zhou et al., 2023) | with | GPT-3.5 | Depth, GPS | 39.2 | 22.3 |
| Egoplan (Ours) | without | GPT-4 | - | **41.2** | **22.5** |

Table 5: Average endpoint error (AEE) results. The lower the number, the closer the image is to the ground truth.

| | Previous flow | Prediction flow |
|---|---|---|
| Habitat 2.0 | 3.30 | **3.09** |
| AI2-THOR | 5.00 | **4.08** |
| VirtualHome | 21.22 | **15.71** |

### 5.3 MOTION PATTERN

As mentioned before, we cannot obtain the optical flow from the current timestep to the next timestep. Therefore, we adopt the VQ-GAN model to predict the current optical flow map. The examples of prediction can be found in Appendix E. We can also found that the VQ-GAN trained on the VH-1.5M dataset can easily generalize to other environments, this is because the optical flow map is a universal feature and does not require the prediction of complex textures.

The average endpoint error (AEE) specifically measures the average distance between two motion vectors at the pixel level. As illustrated in Table 5, the gap between the predicted optical flow map and ground truth is narrower than that between the previous flow map and ground truth (current optical flow map). In addition, the model trained on VirtualHome can still predict optical flow maps in Habitat 2.0 and AI2-THOR (Kolve et al., 2017). This confirms the effectiveness and generalization of the VQ-GAN.

### 5.4 GENERALIZATION

To assess the generalization of our method, we also evaluate its performance in a new household environment. In more detail, we choose Habitat 2.0 due to its high-fidelity scenes compared with other simulators, such as AI2-THOR. However, Habitat 2.0 does not provide any inter-frame regarding manipulation skills, which is unrealistic. Therefore, we only carry out experiments on navigation tasks.

To enhance usability, we use the pre-trained optical flow model, RAFT (Teed & Deng, 2020), to calculate the optical flow for the previous step since the optical flow cannot be directly obtained. The RAFT results are shown in the last 2 columns of Figure 7. Since VQ-GAN has demonstrated some degree of generalization ability to Habitat 2.0 in Section 5.3, we can predict the motion pattern of the new environment. The remaining task is to transfer the visual style to a new environment, and we adopt LoRA to fine-tune the dynamic model. As shown in Figure 5, we successfully perform style transfer with a small amount of data (tens of samples), and the results with LoRA are closer to real scene images compared to those without LoRA visually.

Table 4 presents the success rate (SR) of LLM-based methods on the HM3D ObjectNav task (Yadav et al., 2023), where our method demonstrates strong generalization with a high SR. Notably, our approach surpasses existing LLM-based methods for the first time, achieving a $+4.5\%$ improvement in SR compared to PixelNav (Cai et al., 2023), which navigates to LLM-deduced points. Furthermore, when compared to mapping-based methods employing LLM-guided frontier exploration, our method shows improvements of $+6.0\%$ against L3MVN (Yu et al., 2023) and $+2.0\%$ against ESC (Zhou et al., 2023).

## 6 CONCLUSION AND LIMITATIONS

This paper introduces EgoPlan, an embodied agent that utilizes an LMM as a one-step planner and a Text2image model as a dynamic model for long-horizon tasks. We demonstrate EgoPlan's capacity for high-quality image generation, precise optical flow prediction, and promising decision-making. Notably, we have shown its generalization capabilities across diverse environments. It is important to acknowledge a current limitation: EgoPlan employs encapsulated skills as actions, precluding direct low-level control (e.g., joint positions), which remains a subject for future research.

## 7 ETHICS STATEMENT

This work adheres to the ICLR Code of Ethics. In this study, no human subjects or animal experimentation was involved. All datasets used, including VH-1.5M Dataset, were sourced in compliance with relevant usage guidelines, ensuring no violation of privacy. We have taken care to avoid any biases or discriminatory outcomes in our research process. No personally identifiable information was used, and no experiments were conducted that could raise privacy or security concerns. We are committed to maintaining transparency and integrity throughout the research process.

## 8 REPRODUCIBILITY STATEMENT

We have made every effort to ensure that the results presented in this paper are reproducible. The experimental setup, including training steps, model configurations, and hardware details, is described in detail in the paper.

Additionally, benchmarks such as Virtualhome and Habitat2.0, are publicly available, ensuring consistent and reproducible evaluation results.

We believe these measures will enable other researchers to reproduce our work and further advance the field.

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

# APPENDIX

## A  LLM USAGE

Large Language Models (LLMs) were used to aid in the writing and polishing of the manuscript. Specifically, we used an LLM to assist in refining the language, improving readability, and ensuring clarity in various sections of the paper. The model helped with tasks such as sentence rephrasing, grammar checking, and enhancing the overall flow of the text.

It is important to note that the LLM was not involved in the ideation, research methodology, or experimental design. All research concepts, ideas, and analyses were developed and conducted by the authors. The contributions of the LLM were solely focused on improving the linguistic quality of the paper, with no involvement in the scientific content or data analysis.

The authors take full responsibility for the content of the manuscript, including any text generated or polished by the LLM. We have ensured that the LLM-generated text adheres to ethical guidelines and does not contribute to plagiarism or scientific misconduct.

## B  RELATED WORK

### B.1  DIFFUSION MODEL

The diffusion model (Ho et al., 2020; Song et al., 2022) has been extensively studied in the field of image generation (Dhariwal & Nichol, 2021; Ho et al., 2021; Rombach et al., 2022) and image editing (Gal et al., 2022; Hertz et al., 2022; Meng et al., 2022). Diffusion models can achieve a high degree of control during the image generation. In more detail, InstructPix2Pix (InstructP2P) (Brooks et al., 2023) trains a conditional diffusion model that, given an input image and text instruction for how to edit it, generates the edited image. ControlNet (Zhang et al., 2023b) is widely used to control the style of the generated image by using various forms of prior information, such as edge information and segmentation. By adding LoRA or adapter (Houlsby et al., 2019) modules to the network, the model trained on one data distribution can also be transferred to other data distributions (different visual styles) through a few picture examples. The images produced by current diffusion models are of very high quality, highly realistic, and easily controllable. It prompts various fields to consider using these generated images to assist in accomplishing other tasks. Our paper adopts the diffusion model to generate task subgoals and predict the image of the next state for decision-making.

### B.2  DYNAMIC MODEL AND WORLD MODEL FOR DECISION-MAKING

In the works of using world model for long-range mission planning, the Dreamer series (Hafner et al., 2020; 2022; 2024) models environmental dynamics in latent space to predict future states within gaming contexts, enabling agents to learn tasks through imagination and reducing the number of interactions needed for effective learning. However, as these world models are developed in latent space rather than pixel space, they often struggle to generalize to unseen tasks and environments. A world model constructed in pixel space may offer improved generalization capabilities. Recent studies have sought to address how to learn world models from large-scale video datasets (Liu et al., 2024). In Genie (Bruce et al., 2024), researchers utilize a latent action representation, though their focus primarily revolves around 2D platform video games or simple robotic actions. By meticulously orchestrating rich data across various dimensions, UniSim (Yang et al., 2023) simulates realistic visual experiences in response to actions performed by humans, robots, and other interactive agents. Overall, the applications of world models extend beyond gaming and robotics. For instance, in Escontrela et al. (2024), frame-by-frame video prediction is employed as a mechanism for providing rewards in reinforcement learning. DynaLang (Lin et al., 2023) explores the integration of language prediction as an element of the world model, enabling the training of multimodal world models using datasets that lack explicit actions or rewards. In DynaLang, the representation is shared between vision and language within the world model.

### B.3 Embodied Agent with LMMs

The successful integration of language as a semantically rich input for interactive decision-making underscores the pivotal role of LMMs in facilitating interaction and decision-making processes (Abramson et al., 2020; Karamcheti et al., 2022; Li et al., 2022). LMMs have also been employed across various environments to support robot navigation (Parisi et al., 2022; Hong et al., 2021; Majumdar et al., 2020) and manipulation tasks (Jiang et al., 2022; Ren et al., 2023; Karamcheti et al., 2022). Recently, numerous approaches have emerged that leverage LMMs to enhance the planning and reasoning capabilities of embodied agents. For instance, SayCan (Ahn et al., 2022) evaluates the affordance of potential actions by combining their probabilities derived from LMMs with a value function. (Zeng et al., 2022) integrate a language and multimodal model (LMM) with a visual-language model and a pre-trained language-conditioned policy (Shridhar et al., 2022) to facilitate open vocabulary robotic tasks. Similarly, Huang et al. (2022a) illustrate that LMMs can be effectively utilized for planning and executing simple household tasks, grounding LMM-generated actions by comparing their embeddings with a predefined list of acceptable actions. To incorporate environmental feedback, Inner Monologue (Huang et al., 2022b) enhances SayCan through a closed-loop principle. This principle is further employed in related works such as (Yao et al., 2023; Huang et al., 2022b; Kim et al., 2024; Singh et al., 2023; Liang et al., 2023; Shinn et al., 2023; Wang et al., 2023c) to continuously monitor agent behaviors and refine plans accordingly for tasks in domains like computer automation and Minecraft. Furthermore, there are methods that prompt language and multimodal models (LMMs) to generate temporally abstracted actions (Zheng et al., 2023). Dasgupta et al. (2023) utilize the LMM as both a planner and a success detector for an agent, with their actor module requiring pre-training using reinforcement learning to enable the agent to adhere to natural language instructions. While these studies yield impressive results, they are heavily dependent on the inherent capabilities of powerful LMMs, such as GPT-4 and PaLM (Chowdhery et al., 2023), which presents challenges when attempting to apply these approaches to smaller LMMs with limited reasoning abilities, such as LLaMA-7B. GLAM (Carta et al., 2023) employs RL fine-tuning to achieve functional grounding of LLMs and LMMs. However, their focus is primarily on simple primitive actions (e.g., turn left, turn right, go forward) evaluated within toy environments, such as BabyAI (Chevalier-Boisvert et al., 2018), using a significantly smaller encoder-decoder LMM, Flan-T5-780M. These primitive actions possess a similar token count and lack substantial semantic meaning, which leads to an underutilization of LMM capabilities. Consequently, they fail to adequately explore the effects of prompt design and address the imbalance within the action space, resulting in additional instability and reduced robustness.

## C Details of Virtualhome tasks

We conducted experiments to evaluate the decision-making ability of all methods in the VirtualHome environment. In total, we investigated 12 complex tasks, with detailed instructions and reference subtasks steps for each task as follows:

Listing 1: Instructions and subtasks.

```
<$one-house instructions$>

1. take and place: take the bread from the toaster and place it on the
    plate on the table
steps: (a). walk to the toaster
    (b). grab the bread
    (c). walk to the table
    (d). place the bread on the plate
2. take and put1: take the apple from the table and put it in the
    microwave
steps: (a). walk to the table
    (b). grab the apple
    (c). walk to the microwave
    (d). open the microwave (if the microwave is closed)
    (e). put the apple in the microwave
3. take and put2: take the book from the table and put it on the
    bookshelf
steps: (a). walk to the table
```

```
    (b). take the book
    (c). grab the book
    (d). walk to the bookshelf
    (e). put the book on the bookshelf
4. take and drink: take the water glass from the table and drink from it
steps: (a). walk to the table
    (b). take the water glass
    (c). drink the water glass
5. turn on sit: turn on the TV and sit down
steps: (a). walk to the TV
    (b). turn on the TV
    (c). walk to the chair
    (d). sit down
6. put apple: Put an apple that is on the table into the bookshelf
steps: (a). walk to the table
    (b). grab the apple
    (c). walk to the bookshelf
    (d). put the apple on the bookshelf

<$two-houses instructions$>

7. take and place2: take the frying pan from the counter and place it in
    the sink
steps: (a). walk to the counter
    (b). grab the frying pan
    (c). walk through the door
    (d). walk to the sink
    (e). place frying pan in the sink
8. take and place3: take the condiment shaker from the bookshelf and
    place it on the table
steps: (a). walk to the bookshelf
    (b). grab the condiment shaker
    (c). walk through the door
    (d). walk to the table
    (e). place condiment shaker on the table
9. take and put3: take the salmon on top of the microwave and put it in
    the fridge
steps: (a). walk to the microwave
    (b). grab the salmon
    (c). walk through the door
    (d). walk to the fridge
    (e). open the fridge (if the fridge is closed)
    (f). put salmon in the fridge
10. take open and put: take the pie on the table and warm it using the
    stove
steps: (a). walk to the table
    (b). grab the pie
    (c). walk through the door
    (d). walk to the stove
    (e). put pie on the stove
    (f). switch on the stove
11. take put and open: put the sponge in the sink and wet it by switching
    on the faucet
steps: (a). walk to the sponge
    (b). grab the sponge
    (c). walk through the door
    (d). walk to the sink
    (e). put sponge in the sink
    (f). switching on the faucet
12. take and put4: take the condiment bottle from the kitchen table and
    put it on the plate
steps: (a). walk through the door
    (b). walk to the kitchen table
    (c). grab the condiment bottle
    (d). walk to the plate
```

```
(e). put pie on the stove
(f). switch on the stove
```

In terms of the average task completion length, since we want to prove the effectiveness of our pipeline in long-term planning, we draw on the metric of calvin benchmark (Mees et al., 2022), where the next subtask is executed after the completion of the previous subtask, that is, the completion of the next subtask is conditional on the completion of the previous subtask. This index represents the average number of subtasks that each pipeline can complete after 100 repeated experiments of each task, which measures the long-term planning ability of the pipeline. One repeated experiment represents an **episode**. Since the virtualhome emulator can return instructions on whether the task was successfully executed, our evaluation is automated to calculate the success rate.

## D    DETAILS OF VH-1.5M'S TEXT ACTIONS

We automatically collected the dataset in the order of action category to object. Firstly, 50 different indoor environments are randomly initialized as House1-50, and then the action types (such as put and walk to) are specified. Under each action type, items in the house are randomly selected as the imposed objects of the action. Such commands are executed in the VirtualHome simulator to form a trajectory in the dataset.

The dataset includes a wide range of action sequences, each meticulously annotated with corresponding text actions. These text actions are crucial for providing contextual information that aligns visual actions with natural language descriptions. Below, we detail the process and structure used to generate the text actions for each action sequence in the dataset.

The generation of text actions for VH-1.5M involves a systematic and automated process. This process ensures consistency and variety in the text actions, which are essential for robust training and evaluation in vision-and-language tasks. The key steps in this process are as follows:

**Verb Selection:** A list of verbs related to various actions (e.g., "walk through," "close," "drink") is predefined. For each identified action sequence directory, a verb is randomly selected from the relevant list. This selection ensures a diverse representation of actions.

**Object Name Extraction:** Each directory represents the object acted upon, which signifies the object affected by the action. However, if the action does not involve an object, such as "walk through" or "turn left," no extraction is necessary.

**Phrase Construction:** Two types of phrases are constructed for each action sequence:

Next Timestep Phrase: Describes the immediate next action in the sequence. For example, "next timestep: redeposit the plate".

Goal State Phrase: Describes the intended final action or goal of the sequence. For example, "the goal state: redeposit plate".

**Prompt File Creation:** The constructed phrases are saved in a prompt json file within the respective action sequence directory. This JSON file contains two keys: "next" and "goal," corresponding to the next timestep phrase and goal state phrase, respectively.

### D.1    MORE EXAMPLES OF THE SAMPLES

We give some samples in the sequence of the task, which are shown in Figure 6. Note that samples in one sequence are arranged in chronological order, with the timestep increasing from top to bottom.

## E    MORE EXAMPLES OF GENERATING IMAGES

More examples of generated images from EgoPlan can be seen in Figure 8. Each line represents a task, and the task prompts are, in order: "capture the chicken", "grasp juice", "grasp the hairproduct", "open the cabinet", "open the microwave", "go left", "make a left", "make a left-hand turn", "make a right", "turn right", "turn to the right", "walk straight ahead".

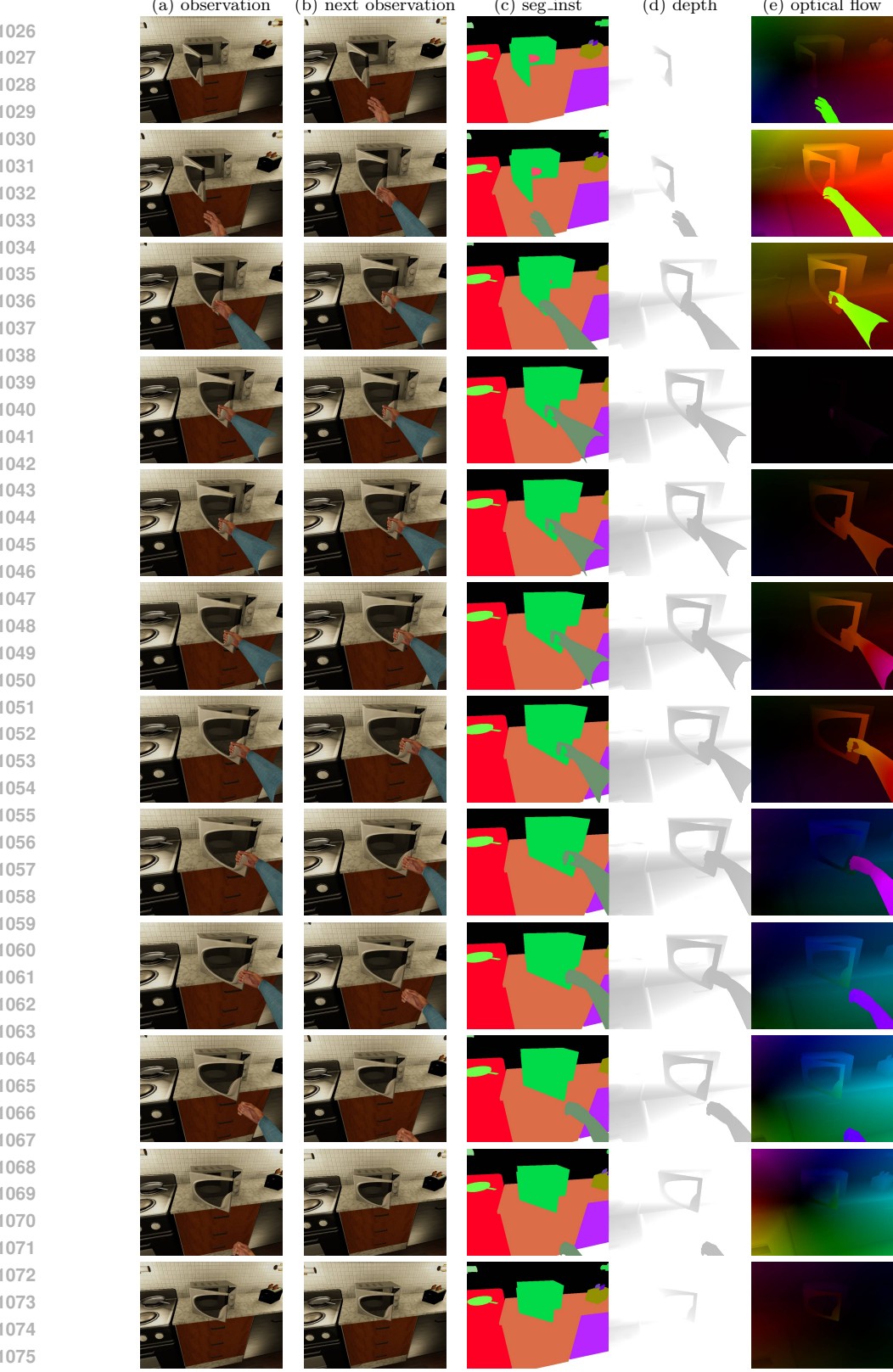

Figure 6: Samples in the sequence of closing the microwave.

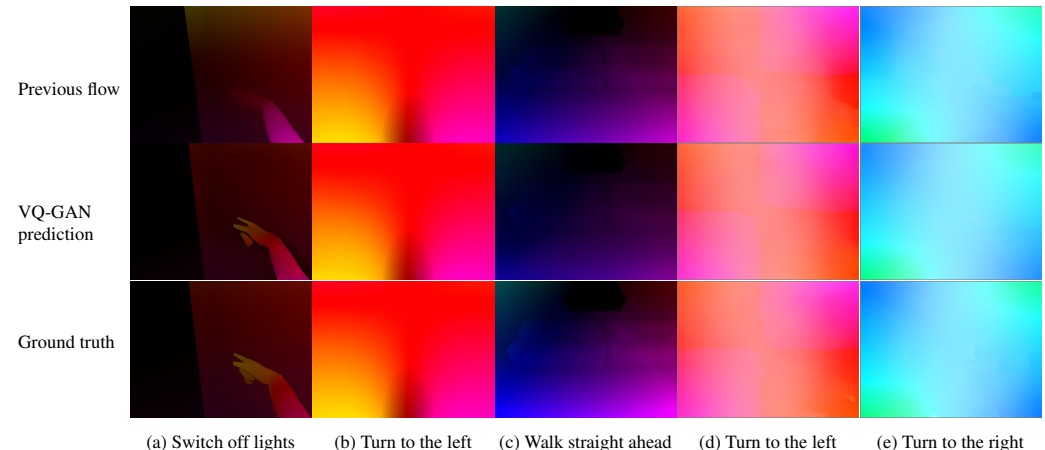

(a) Switch off lights    (b) Turn to the left    (c) Walk straight ahead    (d) Turn to the left    (e) Turn to the right

Figure 7: Examples of optical flow prediction by VQ-GAN. The first 3 columns are optical flow from the VirthualHome environment. The last 2 columns are optical flow from the Habitat2.0 environment.

As illustrated in Figure 7a and 7c, the quality of optical flow prediction for details is promising. Furthermore, as demonstrated in Figure 7d and 7e, the VQ-GAN trained on the VH-1.5M dataset can easily generalize to other environments.

Table 6: User study for the subgoal generation. The user score is the percentage of images that users consider to meet the criteria out of the total 1000 images.

|                    | Close | Drink   | Grab       | Open      | Put back     | Put in |
|--------------------|-------|---------|------------|-----------|--------------|--------|
| Mean user score(%) | 66.5  | 71.75   | 55         | 66.375    | 62.125       | 64.625 |
|                    | Sit   | Stand up| Switch off | Switch on | Walk through |        |
| Mean user score(%) | 79.875| 78.75   | 73.375     | 77.875    | 79           |        |

## F  USER STUDY OF SUBGOAL IMAGE GENERATION

We also conduct a user study on the image generation of the subgoal. A total of 8 users evaluated whether the generated image met the criteria of the subgoal described in the text. Each user evaluates 100 generated images for each action, and the evaluation results are shown in Table 6. The results show that most of the generated subgoal images can represent the meaning of the text subgoals. More examples of generating figures can be seen in Figure 9. It is worth mentioning that after our dataset and training, the subgoal prediction model exhibits certain scene understanding ability. For example, the "power up the lightswitch" subgoal illumines objects (like walls) in a room in the view scene, which is interesting compared to previous work where image generation of subgoals helps decision making.

## G  PROMPT OF TASK-DECOMPOSITION AND LOW-LEVEL ACTION SELECTION

We conducted experiments with detailed query prompt for each task as follows:

Listing 2: query for action selection.

```
Start working. The picture of what you can see has been given above, the
    picture is what you see from the first person perspective as the
    person in the room. Analyze the scene and all the items in the
    picture to make a task plan to complete the instruction.
The instruction is as follows:
"""
{"instruction": [INSTRUCTION]}
"""
The history is as follows:
```

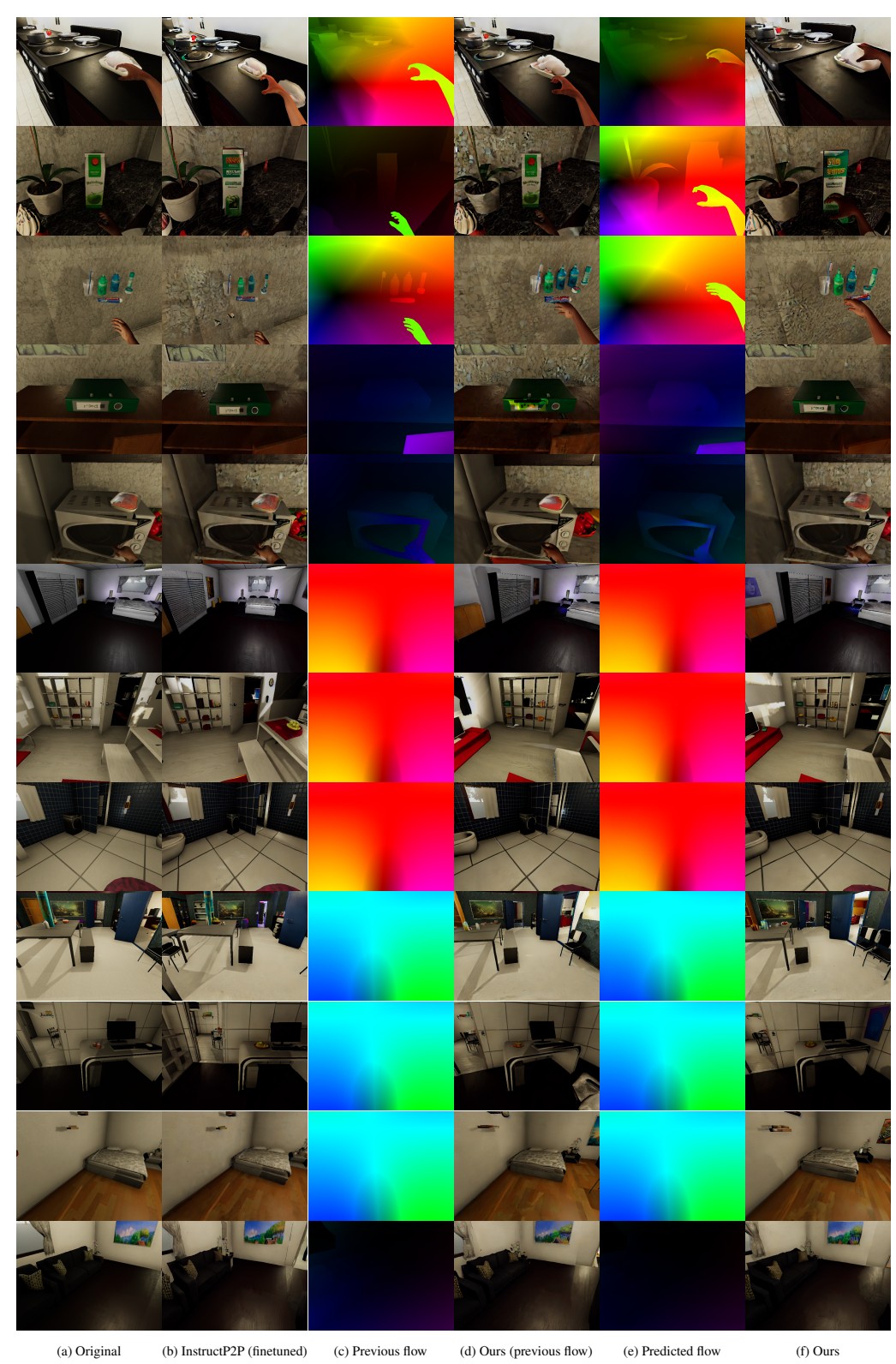

(a) Original     (b) InstructP2P (finetuned)     (c) Previous flow     (d) Ours (previous flow)     (e) Predicted flow     (f) Ours

Figure 8: Examples of the generated image of the EgoPlan in VirtualHome. We can find that in some hand reconstruction and direction understanding scenes, the model without introducing optical flow prior information often performs poorly.

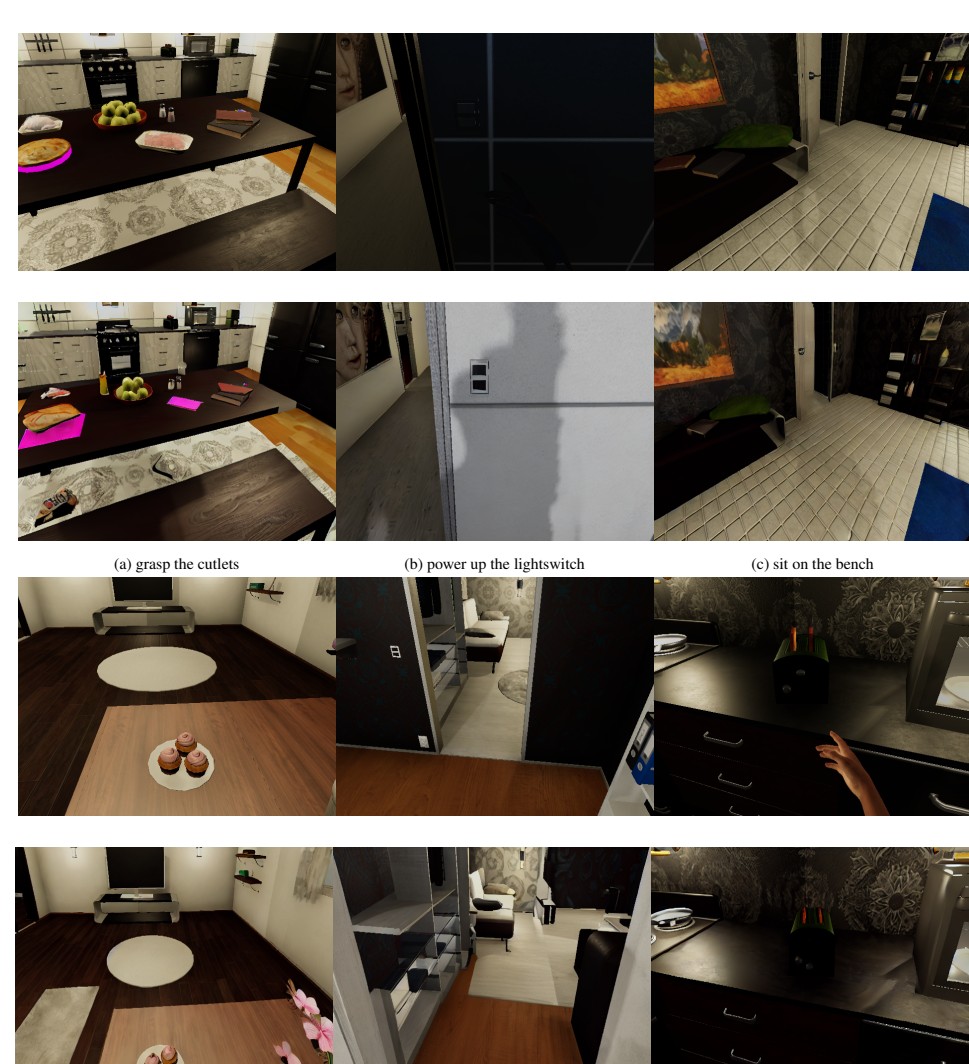

(a) grasp the cutlets     (b) power up the lightswitch     (c) sit on the bench

(d) stand from sofa     (e) walk through     (f) power up toaster

Figure 9: Examples of the generated image subgoals. The first and third rows is the original image, and the second and forth rows is the image subgoal generated based on the text subgoal.

```
"""
{"history": [HISTORY]}
"""
You return should follow these rules:
1. Make sure you provide 4 lines of output each time, the first line is
   the ["Preoperation"] and the secondline is the ["Postoperation"] of
   the action to be taken in the current task plan, and the third line
   is the action to be taken in the plan, which is the ["task_sequence
   "]. The fourth line is the natural language expression of the action
   taken, namely ["step_instructions"]. When output the answer, do not
   attach "step_instructions", "task_sequence", etc.
2. In addition to these, other problem such as input images is too dark
   and historical actions is empty, please DO NOT output.
3. Make sure that element of the ["step_instructions"] explains
   corresponding element of the ["task_sequence"]. That is, the fourth
   line explains the third line.
4. DO NOT USE undefined verbs. USE ONLY verbs in "HUMAN ACTION LIST".
5. The first line and the second line are detailed explanation of the
   forth line. For the task in the forth line, it must be explained in
   two parts: ["Preoperation"] and ["Postoperation"] in the first and
```

```
     second line, separately represents the action state of the agent and
     item before and after the execution of the task.
6. Look carefully at the output examples provided. DO NOT use any strings
    or spaces at the end of sentences. Never left ',' at the end of the
    sentences. STRICTLY ENSURE that the output is always four lines long,
    with no blank lines.
7. The environment given is a picture that you see from the first person
    perspective as the person in the room. Analyze the scene and all the
    items in the picture to make a task plan. If you see a picture that
    is all balck, this means there has been no task planning or execution
    before, please give a general task plan, but BE SURE to stick to the
    output format shown earlier.
8. When selecting each action for task planning, carefully think about
    the function of the action in terms of the two parts ["Preconditions
    "] and ["Postconditions"] after the action, where ["Preconditions"]
    represents the state of the environment before the action is executed
    , and ["Postconditions"] represents the state of the environment
    after the execution, after which the planning is carried out.
9. All sentences you output should NOT be double-quoted.
10. Please strictly correspond to the actions and items in the
    instructions, please strictly keep the spelling of the items, for
    multi-word items, please do not add connection symbols between words,
    for items composed of single-word, please do not split the word.
11. The history is a string that records the actions performed in the
    past few steps, separated by " ". Please plan what action to perform
    at this step based on the historical actions, instructions and the
    current picture.
12. Make sure that you output a consistent manipultation as a human. For
    example, grasping an object should not occur in successive steps.
    Consider whether the current action is simliar to the last action in
    the history. DO NOT output same two actions in row.
13. Every time you do task planning, you should consider whether the
    historical action in history and the current action have completed
    the instruction, and if so, output "Stop()" in time.
Adhere to the output format I defined above. Follow the nine rules. Think
    step by step.
```

We conducted experiments with detailed environment, role of LMM, action function, few-shot output example prompt for each task as follows:

Listing 3: prompt for environment.

```
[user]
Information about environments and objects are given as a picture that
    can be seen from the first person perspective. The picture will be
    given in the example latter.
-------------------------------------------------
The texts above are part of the overall instruction. Do not start working
    yet:
[assistant]
Understood. I will wait for further instructions before starting to work.
```

Listing 4: prompt for role of LMM.

```
[user]
You are an excellent interpreter of human instructions for household
    tasks. Given an instruction and information about the working
    environment, you break it down into a sequence of human actions.
Please do not begin working until I say "Start working." Instead, simply
    output the message "Waiting for next input." Understood?
[assistant]
Waiting for next input.
```

Listing 5: prompt for explanation of action function.

```
[user]
Necessary and sufficient human actions are defined as follows:
"""
"HUMAN ACTION LIST"

Walk(arg1): Walks some distance towards a room or object.
Preconditions: If the environment represented by picture doesn't have the
    obj1 for the task decomposition you did to perform the action, add a
    subtask of Walk(obj1) before the task.

Grab(arg1): Grabs an object.
Preconditions: The object1 property is grabbable (except water). The
    character is close to obj1. obj1 is reachable (not inside a closed
    container). The character has at least one free hand.
Postconditions: Adds a directed edge: character holds_rh or hold_lh, obj1
    . obj1 is no longer on a surface or inside a container.

Open(arg1): Opens an object.
Preconditions: The obj1 property is IS_OPENABLE and the state is closed.
    The character is close to obj1. obj1 is reachable (not inside a
    closed container). The character has at least one free hand.
Postconditions: The obj1 state is open.

Close(arg1): Closes an object.
Preconditions: The obj1 property is IS_OPENABLE and the state is open.
    The character is close to obj1. obj1 is reachable (not inside a
    closed container). The character has at least one free hand.
Postconditions: The obj1 state is closed.

Put(arg1, arg2): Puts an object on another object.
Preconditions: The character holds_lh obj1 or character holds_rh obj1.
    The character is close to obj2.
Postconditions: Removes directed edges: character holds_lh obj1 or
    character holds_rh obj1. Adds directed edges: obj1 on obj2.

PutIn(arg1, arg2): Puts an object inside another object that is OPENABLE,
    such as stove and microwave.
Preconditions: The character holds_lh obj1 or character holds_rh obj1.
    The character is close to obj2. obj2 is not closed. If obj2 is closed
    , The character should open obj2 first and put obj1 in obj2.
Postconditions: Removes directed edges: character holds_lh obj1 or
    character holds_rh obj1. Adds directed edges: obj1 inside obj2.

SwitchOn(arg1): Turns an object on.
Preconditions: The obj1 has the property "switch." The obj1 state is off.
     The character is close to obj1.
Postconditions: The obj1 state is on.

SwitchOff(arg1): Turns an object off.
Preconditions: The obj1 has the property "switch." The obj1 state is on.
    The character is close to obj1.
Postconditions: The obj1 state is off.

Drink(arg1): Drinks from an object.
Preconditions: The obj1 property is drinkable or recipient. The character
     is close to obj1.

Sit(arg1): Sit down on an object.
Preconditions: The obj1 property is sittable. The character is close to
    obj1.

Stop(): The instruction can end the task sequence after the completion of
     the task by the planned instruction.
```

```
Preconditions: After the instruction is decomposed into a series of tasks
    , these tasks fulfill all the requirements of the instruction to be
    executed in order, that is, the instruction is completed in the
    history.
"""
------------------------------------------------
The texts above are part of the overall instruction. Do not start working
    yet:
[assistant]
Waiting for next input.
```

Listing 6: prompt for output example.

```
[user]
I will give you some examples of the input and the output you will
    generate.
Example 1:
"""
- Input:
The picture of what you can see has been given above.
"instruction": "take the salmon on top of the microwave and put it in the
     fridge"
"history": ""
- Output:
The microwave where the salmon is located appears to be distant or out of
     reach, and I need to approach it to interact with it.
I am now close enough to the microwave to interact with it, specifically
    to reach the salmon.
Walk(<microwave>)
Walk towards the microwave to reach the salmon on top.
"""
------------------------------------------------
Example 2:
"""
- Input:
The picture of what you can see has been given above.
"instruction": "take the salmon on top of the microwave and put it in the
     fridge"
"history": "Walk(<microwave>)"
- Output:
The salmon is on top of the microwave and within reach. I have at least
    one free hand to grab it.
I am now holding the salmon, which is no longer on the microwave.
Grab(<salmon>)
Grab the salmon from the top of the microwave
"""
------------------------------------------------
Example 3:
"""
- Input:
The picture of what you can see has been given above.
"instruction": "take the salmon on top of the microwave and put it in the
     fridge"
"history": "Walk(<microwave>)""Grab(<salmon>)"
- Output:
The fridge appears to be distant or out of reach, and I need to approach
    it to interact with it.
I am now close enough to the fridge to put the salmon inside.
Walk(<fridge>)
Walk to the fridge with the salmon
"""
------------------------------------------------
Example 4:
"""
- Input:
```

```
The picture of what you can see has been given above.
"instruction": "take the salmon on top of the microwave and put it in the
    fridge"
"history": "Walk(<microwave>)""Grab(<salmon>)""Walk(<fridge>)"
- Output:
Before I can put the salmon inside, the fridge must be open.
The fridge is now open, and I can place items inside.
Open(<fridge>)
Open the fridge
"""
-------------------------------------------------
Example 5:
"""
- Input:
The picture of what you can see has been given above.
"instruction": "take the salmon on top of the microwave and put it in the
    fridge"
"history": "Walk(<microwave>)""Grab(<salmon>)""Walk(<fridge>)""Open(<
    fridge>)"
- Output:
I hold the salmon. I am close to the fridge which is now open.
The salmon is now inside the fridge, and my hands are free.
PutIn(<salmon>, <fridge>)
Put the salmon in the fridge
"""
-------------------------------------------------
Example 6:
"""
- Input:
The picture of what you can see has been given above.
"instruction": "take the salmon on top of the microwave and put it in the
    fridge"
"history": "Walk(<microwave>)""Grab(<salmon>)""Walk(<fridge>)""Open(<
    fridge>)""PutIn(<salmon>, <fridge>)"
- Output:
After placing the salmon inside, the fridge remains open.
The fridge is now closed, securing the salmon inside.
Close(<fridge>)
Close the fridge door
"""
-------------------------------------------------
Example 7:
"""
- Input:
The picture of what you can see has been given above.
"instruction": "take the salmon on top of the microwave and put it in the
    fridge"
"history": "Grab(<salmon>)""Walk(<fridge>)""Open(<fridge>)""PutIn(<salmon
    >, <fridge>)""Close(<fridge>)"
- Output:
I take the salmon on top of the microwave and put it in the fridge.
The instruction has been finished.
Stop()
Complete the instruction and stop the task planning
"""
-------------------------------------------------
The texts above are part of the overall instruction. Do not start working
    yet:
[assistant]
Waiting for next input.
```

Listing 7: prompt for output format.

```
[user]
```

```
You divide the actions given in the text into detailed robot actions and
    put them together as a python dictionary.
The dictionary has three keys.
"""
- dictionary["task_cohesion"]: A dictionary containing information about
    the robot's actions that have been split up.
- dictionary["instruction_summary"]: contains a brief summary of the
    given sentence.
"""
Two keys exist in dictionary["task_cohesion"].
"""
- dictionary["task_cohesion"]["task_sequence"]: A dictionary containing
    information about the human's actions that have been split up.
- dictionary["task_cohesion"]["step_instructions"]: contains a brief text
     explaining why this step is necessary.
-------------------------------------------------
The texts above are part of the overall instruction. Do not start working
    yet:
[assistant]
Waiting for next input.
```

## H  TRAJECTORIES OF SELF-REFLECTION IN NAVIGATION TASKS

When executing navigation tasks, the subgoal is "walk to (<somewhere>)" while the underlying actions include "walk forward", "turn left", and "turn right". Accomplishing the navigation task with such a subgoal constitutes a long-horizon composite task. In addressing these types of tasks, we employ a **React+Reflexion** mechanism that leverages previous actions to perform operations such as obstacle avoidance and target searching. For example, we can illustrate trajectories based on these actions. The trajectories under the self-reflection mechanism are shown as follows:

Listing 8: Trajectory of self-reflection.

```
Trial #1
Environment: <observation image>. You are in the middle of a room.
    Looking quickly around you, you see a bed 1, a desk 2, a desk 1, a
    computer 1, a drawer 5, a drawer 4, a drawer 3, a drawer 2, a drawer
    1, a garbagecan 1, a laundryhamper 1, a cabinet 1, a shelf 6, a shelf
     5, a shelf 4, a shelf 3, a shelf 2, and a shelf 1.
Task: walk to computer 1.
> think: To solve the task, I need to find a desk, then find a computer.
> think: First I need to find a computer. From the obsersation image, a
    computer is on desk 1, from the dynamic model output, you should turn
     left to get direction.
> turn left
> success
Environment: <observation image>. You are in the middle of a room.
    Looking quickly around you, you see a bed 1, a desk
2, a desk 1, a computer 1, a drawer 5, a drawer 4, a drawer 3, a drawer
    2, a drawer 1, a garbagecan 1, a
laundryhamper 1, a cabinet 1, a shelf 6, a shelf 5, a shelf 4, a shelf 3,
     a shelf 2, and a shelf 1.
> think: First I need to find a computer. From the obsersation image, a
    computer is on desk 1, from the dynamic model output, you should turn
     left to get direction.
> turn left
> success
Environment: <observation image>. You are in the middle of a room.
    Looking quickly around you, you see a bed 1, a desk
2, a desk 1, a computer 1, a drawer 5, a drawer 4, a drawer 3, a drawer
    2, a drawer 1, a garbagecan 1, a
laundryhamper 1, a cabinet 1, a shelf 5, a shelf 4, a shelf 3, a shelf 2,
     and a shelf 1.
```

```
> think: You are get right direction of the desk, the desk is in the
    middle of the egocentric view.
> walk forward
> Nothing happens.
Environment: <observation image>. You are in the middle of a room.
    Looking quickly around you, you see a bed 1, a desk
2, a desk 1, a computer 1, a drawer 5, a drawer 4, a drawer 3, a drawer
    2, a drawer 1, a garbagecan 1, a
laundryhamper 1, a cabinet 1, a shelf 5, a shelf 4, a shelf 3, a shelf 2,
    and a shelf 1.
> think: You are get right direction of the desk, the desk is in the
    middle of the egocentric view.
> walk forward
> Nothing happens.
Environment: <observation image>. You are in the middle of a room.
    Looking quickly around you, you see a bed 1, a desk
2, a desk 1, a computer 1, a drawer 5, a drawer 4, a drawer 3, a drawer
    2, a drawer 1, a garbagecan 1, a
laundryhamper 1, a cabinet 1, a shelf 5, a shelf 4, a shelf 3, a shelf 2,
    and a shelf 1.
> think: You are get right direction of the desk, the desk is in the
    middle of the egocentric view.
> walk forward
> Nothing happens.
Status: Fail

Reflection: When we locate the computer and identify the correct
    direction, we find that the action to approach the computer is not
    responding. Observing first-person perspective images reveals that
    there is a bed in front, acting as an obstacle that prevents forward
    movement. To get closer to the computer, it is necessary to navigate
    around the bed first.

Trial #2
Environment: <observation image>. You are in the middle of a room.
    Looking quickly around you, you see a bed 1, a desk 2, a desk 1, a
    computer 1, a drawer 5, a drawer 4, a drawer 3, a drawer 2, a drawer
    1, a garbagecan 1, a laundryhamper 1, a cabinet 1, a shelf 6, a shelf
    5, a shelf 4, a shelf 3, a shelf 2, and a shelf 1.
Task: walk to computer 1.
> think: To solve the task, I need to find a desk, then find a computer.
> think: First I need to find a computer. From the obsersation image, a
    computer is on desk 1, from the dynamic model output, you should turn
    right to get around the bed.
> turn right
> success
Environment: <observation image>. You are in the middle of a room.
    Looking quickly around you, you see a bed 1, a desk 2, a desk 1, a
    computer 1, a drawer 5, a drawer 4, a drawer 3, a drawer 2, a drawer
    1, a laundryhamper 1, a cabinet 1, a shelf 6, a shelf 5, a shelf 4, a
    shelf 3, a shelf 2, and a shelf 1, a laundryhamper 2.
> think: First I need to find a computer. From the obsersation image, a
    computer is on desk 1, from the dynamic model output, you should turn
    right to get around the bed.
> turn right
> success
Environment: <observation image>. You are in the middle of a room.
    Looking quickly around you, you see a bed 1, a desk 2, a desk 1, a
    computer 1, a drawer 5, a drawer 4, a drawer 3, a drawer 2, a drawer
    1, a laundryhamper 1, a cabinet 1, a shelf 6, a shelf 5, a shelf 4, a
    shelf 3, a shelf 2, and a shelf 1, a laundryhamper 2.
> think: First I need to find a computer. From the obsersation image, a
    computer is on desk 1, from the dynamic model output, you should turn
    right to get around the bed.
> turn right
```

```
> success
Environment: <observation image>. You are in the middle of a room.
    Looking quickly around you, you see a bed 1, a desk 2, a desk 1, a
    computer 1, a drawer 5, a drawer 4, a drawer 3, a drawer 2, a drawer
    1, a laundryhamper 1, a cabinet 1, a shelf 6, a shelf 5, a shelf 4, a
     shelf 3, a shelf 2, and a shelf 1, a laundryhamper 2.
> think: First I need to find a computer. From the obsersation image, a
    computer is on desk 1, currently, there are no obstacles blocking
    your path in front of you. From the dynamic model output, you should
    walk forward to get around the bed.
> walk forward
> success
Environment: <observation image>. You are in the middle of a room.
    Looking quickly around you, you see a bed 1, a desk 2, a desk 1, a
    computer 1, a drawer 5, a drawer 4, a drawer 3, a drawer 2, a drawer
    1, a laundryhamper 1, a cabinet 1, a shelf 6, a shelf 5, a shelf 4, a
     shelf 3, a shelf 2, and a shelf 1, a laundryhamper 2.
> think: First I need to find a computer. From the obsersation image, a
    computer is on desk 1, currently, there are no obstacles blocking
    your path in front of you. From the dynamic model output, you should
    walk forward to get around the bed.
> walk forward
> success
Environment: <observation image>. You are in the middle of a room.
    Looking quickly around you, you see a bed 1, a desk 2, a desk 1, a
    computer 1, a drawer 5, a drawer 4, a drawer 3, a drawer 2, a drawer
    1, a laundryhamper 1, a cabinet 1, a shelf 6, a shelf 5, a shelf 4, a
     shelf 3, a shelf 2, and a shelf 1, a laundryhamper 2.
> think: First I need to find a computer. From the obsersation image, a
    computer is on desk 1, currently, there are no obstacles blocking
    your path in front of you. From the dynamic model output, you should
    walk forward to get around the bed.
> walk forward
> success
Environment: <observation image>. You are in the middle of a room.
    Looking quickly around you, you see a bed 1, a desk 2, a desk 1, a
    computer 1, a drawer 5, a drawer 4, a drawer 3, a drawer 2, a drawer
    1, a laundryhamper 1, a cabinet 1, a shelf 6, a shelf 5, a shelf 4, a
     shelf 3, a shelf 2, and a shelf 1, a laundryhamper 2.
> think: First I need to find a computer. From the obsersation image, a
    computer is on desk 1, currently, there are no obstacles blocking
    your path in front of you. From the dynamic model output, you should
    walk forward to get around the bed.
> walk forward
> success
Environment: <observation image>. You are in the middle of a room.
    Looking quickly around you, you see a bed 1, a desk
2, a desk 1, a computer 1, a drawer 5, a drawer 4, a drawer 3, a drawer
    2, a drawer 1, a garbagecan 1, a
laundryhamper 1, a cabinet 1, a shelf 6, a shelf 5, a shelf 4, a shelf 3,
    a shelf 2, and a shelf 1.
> think: First I need to find a computer. From the obsersation image, a
    computer is on desk 1, from the dynamic model output, you should turn
     left to get direction.
> turn left
> success
Environment: <observation image>. You are in the middle of a room.
    Looking quickly around you, you see a bed 1, a desk
2, a desk 1, a computer 1, a drawer 5, a drawer 4, a drawer 3, a drawer
    2, a drawer 1, a garbagecan 1, a
laundryhamper 1, a cabinet 1, a shelf 6, a shelf 5, a shelf 4, a shelf 3,
    a shelf 2, and a shelf 1.
> think: First I need to find a computer. From the obsersation image, a
    computer is on desk 1, from the dynamic model output, you should turn
     left to get direction.
```

```
> turn left
> success
Environment: <observation image>. You are in the middle of a room.
    Looking quickly around you, you see a bed 1, a desk
2, a desk 1, a computer 1, a drawer 5, a drawer 4, a drawer 3, a drawer
    2, a drawer 1, a garbagecan 1, a
laundryhamper 1, a cabinet 1, a shelf 6, a shelf 5, a shelf 4, a shelf 3,
    a shelf 2, and a shelf 1.
> think: First I need to find a computer. From the obsersation image, a
    computer is on desk 1, from the dynamic model output, you should turn
    left to get direction.
> turn left
> success
Environment: <observation image>. You are in the middle of a room.
    Looking quickly around you, you see a bed 1, a desk
2, a desk 1, a computer 1, a drawer 5, a drawer 4, a drawer 3, a drawer
    2, a drawer 1, a garbagecan 1, a
laundryhamper 1, a cabinet 1, a shelf 6, a shelf 5, a shelf 4, a shelf 3,
    a shelf 2, and a shelf 1.
> think: First I need to find a computer. From the obsersation image, a
    computer is on desk 1, from the dynamic model output, you should turn
    left to get direction.
> turn left
> success
Environment: <observation image>. You are in the middle of a room.
    Looking quickly around you, you see a bed 1, a desk 2, a desk 1, a
    computer 1, a drawer 5, a drawer 4, a drawer 3, a drawer 2, a drawer
    1, a laundryhamper 1, a cabinet 1, a shelf 6, a shelf 5, a shelf 4, a
    shelf 3, a shelf 2, and a shelf 1, a laundryhamper 2.
> think: First I need to find a computer. From the obsersation image, a
    computer is on desk 1. From the dynamic model output, you should walk
    forward to get around the bed.
> walk forward
> success
Environment: <observation image>. You are in the middle of a room.
    Looking quickly around you, you see a bed 1, a desk 2, a desk 1, a
    computer 1, a drawer 5, a drawer 4, a drawer 3, a drawer 2, a drawer
    1, a laundryhamper 1, a cabinet 1, a shelf 6, a shelf 5, a shelf 4, a
    shelf 3, a shelf 2, and a shelf 1, a laundryhamper 2.
> think: First I need to find a computer. From the obsersation image, a
    computer is on desk 1. From the dynamic model output, you should walk
    forward to get around the bed.
> walk forward
> success
Environment: <observation image>. You are in the middle of a room.
    Looking quickly around you, you see a bed 1, a desk 2, a desk 1, a
    computer 1, a drawer 5, a drawer 4, a drawer 3, a drawer 2, a drawer
    1, a laundryhamper 1, a cabinet 1, a shelf 6, a shelf 5, a shelf 4, a
    shelf 3, a shelf 2, and a shelf 1, a laundryhamper 2.
> think: First I need to find a computer. From the obsersation image, a
    computer is on desk 1. From the dynamic model output, you should walk
    forward to get around the bed.
> walk forward
> success
Environment: <observation image>. You are in the middle of a room.
    Looking quickly around you, you see a bed 1, a desk 2, a desk 1, a
    computer 1, a drawer 5, a drawer 4, a drawer 3, a drawer 2, a drawer
    1, a laundryhamper 1, a cabinet 1, a shelf 6, a shelf 5, a shelf 4, a
    shelf 3, a shelf 2, and a shelf 1, a laundryhamper 2.
> think: First I need to find a computer. From the obsersation image, a
    computer is on desk 1. From the dynamic model output, you should walk
    forward to get around the bed.
> walk forward
> success
```

```
Status: Success
```

## I    TRAINING DETAILS

**VQ-GAN** Our model was trained on the VH-1.5M dataset. The training set consists of motion trajectories from the first 49 rooms, while the last room was used as the validation set. Each room contains approximately 30,000 frames of images. The images were normalized and augmented using random cropping and horizontal flipping. We trained the model from scratch, where the input for each frame was the optical flow of the previous frame. The model was tasked with predicting the optical flow of the next frame based on this input. We used a batch size of 12 and trained the model for 50 epochs with an initial learning rate of $3.5 \cdot 10^{-5}$. The training process was conducted on eight NVIDIA A100 GPUs, each with 40GB of memory, and the total training time was approximately four days.

**Instructpix2pix** Our model was trained on the VH-1.5M dataset. The training set consists of motion trajectories from the first 49 rooms, while the last room was used as the validation set. Each room contains approximately 30,000 frames of images. The images were normalized and augmented using random cropping and horizontal flipping. We trained the model from pretraining, where the input for each frame was the previous frame. The model was tasked with predicting the next frame based on this input. We used a batch size of 32 and trained the model for 50000 epochs. We use cosine annealing to drop the learning rate from $10^{-4}$ to $10^{-5}$ for the first 20,000 training rounds. The training process was conducted on eight NVIDIA A100 GPUs, each with 40GB of memory, and the total training time was approximately two days.

**ControlNet** Our model was trained on the VH-1.5M dataset. The training set consists of motion trajectories from the first 49 rooms, while the last room was used as the validation set. Each room contains approximately 30,000 frames of images. The images were normalized and augmented using random cropping and horizontal flipping. We initialize the model weights to 0 then train the model from scratch, where the input for each frame was the optical flow of the previous frame. The model was tasked with predicting the the next frame based on this input. We used a batch size of 24 and trained the model for 80000 epochs. We use cosine annealing to drop the learning rate from $10^{-4}$ to $10^{-5}$ for the first 40,000 training rounds. The training process was conducted on eight NVIDIA A100 GPUs, each with 40GB of memory, and the total training time was approximately four days.

## J    COMPLEXITY ANALYSIS

**In the section, we will discuss why does our pipeline employ one-step planning? This is actually based on striking a banlance between decision accuracy and complexity.** For our problem settings, we use Partially Observable Markov Decision Process (POMDP) (Smallwood & Sondik, 1973) to define the decision making process due to egocentric view is the partial observation for Egoplan agent. When we use our dynamics model to do multi-step prediction, due to the number of possible future states goes up very fast, can we guarantee a significant improvement in decision making (task completion success rate) without an explosive increase in the number of decisions in GPT4V? We calculated the relationship between action decision accuracy (compared to a skillfull human expert) and the number of GPT4V decisions for different dynamic model prediction steps in different tasks (the first six virtualhome tasks), and constructed a statistic $\frac{accuracy}{complexity}$ (the larger statistic indicates the more "effective" and "skillfull" of agent's decision-making assisted by this dynamic model), the results are as shown in the Table 7. These results point out that in some long-horizon tasks with huge changes in perspective, when an agent with egocentric view is performing model predictive control, using some AI agent decision technology, multi-step prediction often brings a lot of computational complexity.

## K    SUCCESS RATE (COMPLETE RATE) OF VIRTUALHOME TASKS

The final success of the long-range tasks on virtualhome tasks is shown in Figure 11. The final completion rate reflects the probability that the agent will reach the end point in the long-term

|              | 1-step | 2-step | 3-step | 4-step |
|--------------|--------|--------|--------|--------|
| take and place | 7.12 | 1.21 | 0.17 | 0.07 |
| take and put1  | 7.01 | 1.34 | 0.25 | 0.03 |
| take and put2  | 6.98 | 1.14 | 0.29 | 0.06 |
| take and drink | 6.74 | 1.06 | 0.31 | 0.09 |
| turn on sit    | 7.32 | 1.31 | 0.35 | 0.02 |
| put apple      | 7.22 | 0.95 | 0.39 | 0.07 |

Table 7: The indicators of decision accuracy and decision numbers $\frac{accuracy}{complexity}$ for dynamic model autoregressive prediction with different number of steps.

task trajectory, and thus reflects the stability of the pipeline. See Figure 10 for a more intuitive representation of Table 4.

# L  REAL WORLD EXPERIMENTS

We conducted experiments in real scenarios. We placed the necessary experimental items in different rooms. In the subsequent version of our work, we will present the experimental environment in more detail. Now, we are presenting the success rate of the task in the real scenario. Compared to the baseline completion rate, Egoplan achieved a higher task completion rate in real scenarios, For our methods, we adopt the diffusion policy (Chi et al., 2023) method as our low-level policy. In real world tasks, we collected approximately 10 trajectories with about 100 frames to fine-tune our world model.

Since the Qwen-2.5-VL-32B model is much smaller than GPT4V and may not have been pre-trained on specialized embodied reasoning datasets, its performance is much worse compared to GPT4V. We also replaced our world model with Cosmos-Predict1-7B-Video2World, which supports both text and video input. We unified our input as the baseline of the current egocentric view (a single frame image) as Cosmos(frame). We found that if we input historical videos into Cosmos as Cosmos(video), then Cosmos would undergo significant improvements and approach the performance of our method (but our method only inputs current observations). We have attached the results in Tables 8 and 9.

Table 8: The number of success on 12 tasks for all the methods. Tasks 1-6 occur inside one room, while tasks 7-12 take place in two rooms. Each task was executed 100 times.

| Task | Qwen-2.5+React | Qwen-2.5-VL | React | Reflexion | VL+P2P | Qwen-2.5-VL+OF |
|------|----------------|-------------|-------|-----------|--------|----------------|
| take and place   | 0 | 0 | 2 | 2 | 4 | 4 |
| take and put1    | 0 | 0 | 2 | 3 | 5 | 4 |
| take and put2    | 0 | 0 | 1 | 2 | 3 | 4 |
| take and drink   | 0 | 0 | 1 | 2 | 4 | 5 |
| turn on sit      | 0 | 0 | 1 | 2 | 4 | 4 |
| put apple        | 0 | 0 | 1 | 1 | 4 | 4 |
| take and place2  | 0 | 0 | 0 | 1 | 3 | 4 |
| take and place3  | 0 | 0 | 1 | 1 | 3 | 4 |
| take and put3    | 0 | 0 | 1 | 2 | 3 | 4 |
| take open and put| 0 | 0 | 1 | 2 | 4 | 4 |
| take put and open| 0 | 0 | 1 | 2 | 4 | 4 |
| take and put4    | 0 | 0 | 0 | 1 | 2 | 3 |

| Task | Cosmos(frame) | GR-SUSIE | GR-MG | Cosmos(video) | Ours(text goal) | Ours(image goal) |
|------|---------------|----------|-------|---------------|-----------------|------------------|
| take and place   | 3 | 4 | 4 | 8  | 6 | 8 |
| take and put1    | 4 | 4 | 3 | 8  | 6 | 9 |
| take and put2    | 3 | 4 | 5 | 10 | 7 | 9 |
| take and drink   | 4 | 4 | 3 | 6  | 6 | 8 |
| turn on sit      | 4 | 4 | 3 | 6  | 5 | 6 |
| put apple        | 3 | 3 | 4 | 7  | 6 | 9 |
| take and place2  | 3 | 3 | 3 | 4  | 3 | 5 |
| take and place3  | 3 | 3 | 3 | 6  | 4 | 8 |
| take and put3    | 3 | 3 | 6 | 4  | 4 | 6 |
| take open and put| 3 | 3 | 4 | 6  | 5 | 5 |
| take put and open| 3 | 4 | 5 | 5  | 5 | 6 |
| take and put4    | 3 | 3 | 4 | 5  | 5 | 4 |

We trained several end-to-end RL/Implicit Learning methods on our dataset. For these methods, we tried our best to uniformly select appropriate model architectures. Here are the various models we chose and their respective effects in Table 10.

Table 9: The number of success on 12 tasks for all the methods. Tasks 1-6 occur inside one room, while tasks 7-12 take place in two rooms. Each task was executed 100 times.

| Task | GPT4+React | GPT4V | React | Reflexion | GPT4V+P2P |
|---|---|---|---|---|---|
| take and place | 0 | 2 | 8 | 10 | 12 |
| take and put1 | 0 | 2 | 7 | 8 | 10 |
| take and put2 | 0 | 3 | 5 | 6 | 8 |
| take and drink | 0 | 1 | 3 | 5 | 8 |
| turn on sit | 0 | 2 | 5 | 6 | 8 |
| put apple | 0 | 3 | 3 | 5 | 10 |
| take and place2 | 0 | 1 | 2 | 3 | 7 |
| take and place3 | 0 | 1 | 3 | 6 | 9 |
| take and put3 | 0 | 0 | 4 | 4 | 9 |
| take open and put | 0 | 1 | 3 | 4 | 7 |
| take put and open | 0 | 2 | 2 | 8 | 12 |
| take and put4 | 0 | 1 | 4 | 6 | 9 |

| Task | GPT4V+OF | SUSIE | GR-MG | Ours(text goal) | Ours(image goal) |
|---|---|---|---|---|---|
| take and place | 14 | 11 | 14 | 16 | 21 |
| take and put1 | 13 | 11 | 12 | 17 | 20 |
| take and put2 | 10 | 9 | 10 | 14 | 18 |
| take and drink | 12 | 9 | 11 | 13 | 15 |
| turn on sit | 13 | 13 | 11 | 12 | 15 |
| put apple | 11 | 11 | 11 | 13 | 17 |
| take and place2 | 9 | 5 | 6 | 9 | 10 |
| take and place3 | 12 | 9 | 11 | 13 | 15 |
| take and put3 | 11 | 9 | 9 | 11 | 13 |
| take open and put | 10 | 12 | 14 | 14 | 15 |
| take put and open | 12 | 12 | 11 | 14 | 14 |
| take and put4 | 12 | 10 | 12 | 14 | 15 |

- **LCBC (Language-Conditioned Behavior Cloning)** For the LCBC baseline, we use the same architecture and hyperparameters as the low-level policy in SUSIE. We encode the language instruction using MUSE (Yinfei Yang et al. Multilingual Universal Sentence Encoder for Semantic Retrieval) and feed it into the ResNet-50 image encoder using FiLM conditioning.

- **PPO** We use ResNet-50 as image emcoder, 3 256-unit MLP layers are used as backbone. When PPO agent accomplish each sub-goal we give `reward = 1`. When PPO agent accomplish goal we give `reward = 10`.

- **GCBC (Goal Conditional Behavior Cloning)** For the GCBC baseline, we need to emphasize that this is actually the method of SUSIE. SUSIE uses `Instructpix2pix` to generate sugoa! and then applies the GCBC method in the downstream model.

- **GCIL (Goal Conditional Imitation Learning)** Observation and goal image are passed into ResNet-50 image encoder. 3 256-unit MLP layers are used as backbone.

Among these methods, we found that three types of information mainly guide the strategy learning: language instructions (LCBC), sub-goals (GCBG, GCIL), and rewards (PPO). Based on the results, the preliminary conclusion we can draw is that sub-goals are the most useful for guiding the learning of strategies.

Table 10: The number of success on 12 tasks for all the methods. Tasks 1-6 occur inside one room, while tasks 7-12 take place in two rooms. Each task was executed 100 times.

| | LCBC | GCBC(SUISIE) | GCIL | PPO | Ours(image goal) |
|---|---|---|---|---|---|
| **take and place** | 10 | 11 | 10 | 8 | **21** |
| **take and put1** | 8 | 11 | 12 | 9 | **20** |
| **take and put2** | 6 | 9 | 13 | 7 | **18** |
| **take and drink** | 5 | 9 | 8 | 10 | **15** |
| **turn on sit** | 5 | 13 | 12 | 10 | **15** |
| **put apple** | 8 | 11 | 10 | 10 | **17** |
| **take and place2** | 3 | 5 | 7 | 7 | **10** |
| **take and place3** | 4 | 9 | 7 | 8 | **15** |
| **take and put3** | 5 | 9 | 7 | 4 | **13** |
| **take open and put** | 5 | 12 | 10 | 9 | **15** |
| **take put and open** | 4 | 12 | 12 | 9 | **14** |
| **take and put4** | 4 | 10 | 14 | 10 | **15** |

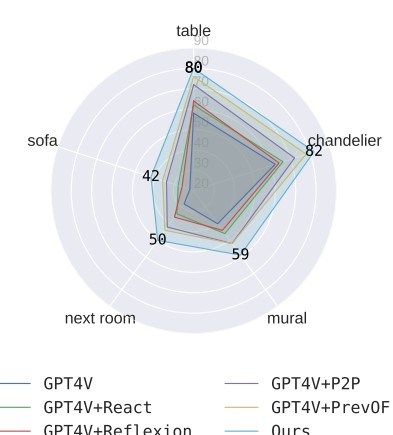

Figure 10: The success rate on 5 navigation tasks for all the methods in Habitat 2.0. GPT4+React is omitted due to its poor performance.

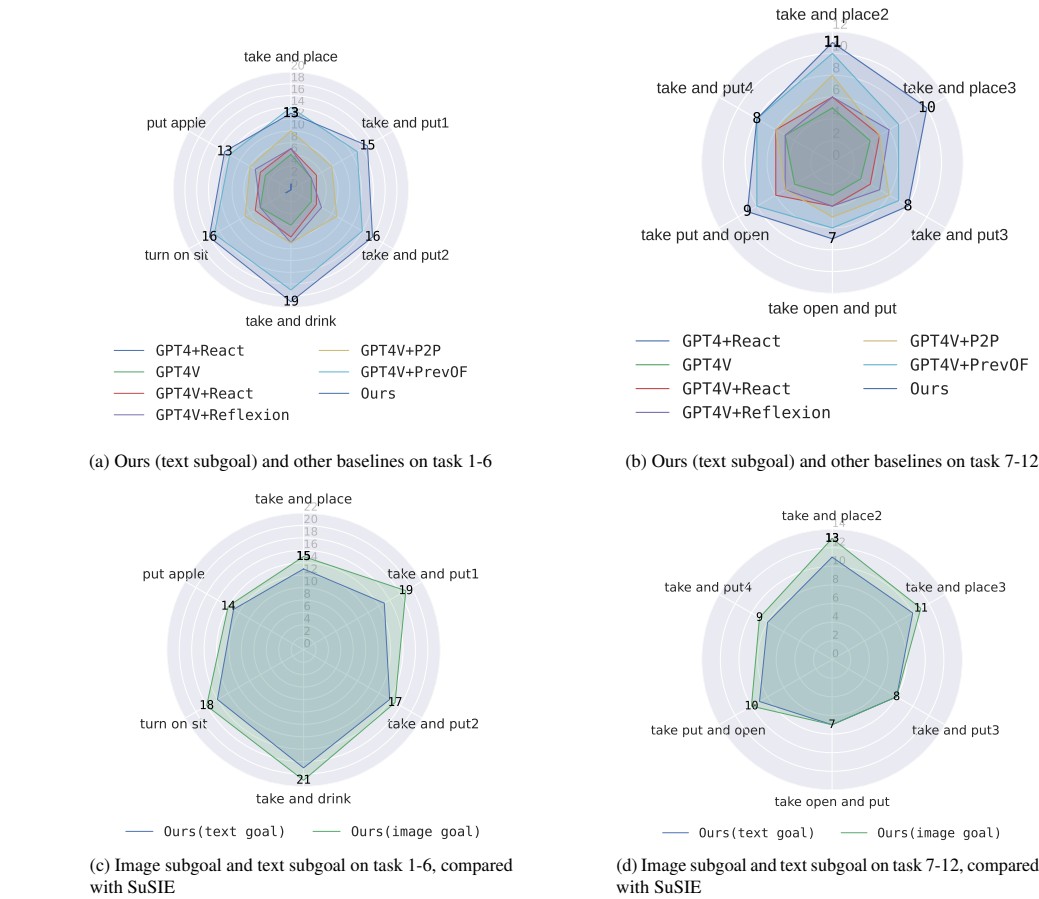

(a) Ours (text subgoal) and other baselines on task 1-6

(b) Ours (text subgoal) and other baselines on task 7-12

(c) Image subgoal and text subgoal on task 1-6, compared with SuSIE

(d) Image subgoal and text subgoal on task 7-12, compared with SuSIE

Figure 11: The success rate on 12 tasks for all the methods. Note that tasks 1-6 occur inside one room, while tasks 7-12 take place in two rooms.

