# OpenReview forum: "EgoPlan: Towards Effective Embodied Agents via Egocentric Planning"
_ICLR.cc/2026/Conference — ICLR 2026 Conference Withdrawn Submission_

### Official Review · Reviewer_M47a · 2025-10-30

**Soundness:** 2
**Presentation:** 3
**Contribution:** 2
**Rating:** 2
**Confidence:** 4

**Summary:**

EgoPlan presents a two-stage pipeline for long-horizon embodied planning from egocentric RGB input.

A diffusion-based one-step dynamics model predicts the next observation conditioned on the current frame, a discrete high-level action, and an explicitly estimated optical-flow field injected via ControlNet. The model is further adapted to new domains with lightweight LoRA fine-tuning.

A large multimodal model (GPT-4V) both decomposes high-level language instructions into sub-goals and performs one-step look-ahead planning: it scores the synthetic next-frame candidates and chooses the action that best progresses toward the current sub-goal, aided by ReAct/Reflexion prompting.

The authors contribute VH-1.5M, a 1.5-million-step egocentric dataset, and report improvements in image quality (FID 1.06 → 0.82) and task progress on 12 VirtualHome scenarios, as well as a 41.2 % zero-shot success rate on HM3D ObjectNav.

**Strengths:**

1.  The method is described in sufficient implementation detail (datasets, loss functions, prompts). Ablations on optical flow and LoRA give partial evidence of each component's value;
2.  Writing is mostly clear;
3.  Long-horizon, partially observable, household tasks are an important and challenging domain; the VH-1.5M dataset should be useful for future work.

**Weaknesses:**

The motivation of the porposed idea is to connect LMMs to the physical world. To address this problem, the authors use a video generation model to predict the future videos given discrete actions. And let the LMM select the best one. My major concerns are as follows:
1.  Action granularity: The action set mixes low-level motions (e.g., “move forward”) with higher-level skills (e.g., “grab object”), making it unclear how well the approach would scale to real robotic settings where precise control and continuous motions are required.
2.  Comparative evidence: In Table 3 the absolute performance remains low and is close to or below simpler baselines.
3.  There are also several previous that use optical flow as condition to generate future videos, for example:

[1] OnlyFlow: Optical Flow based Motion Conditioning for Video Diffusion Models

[2] FloVD: Optical Flow Meets Video Diffusion Model for Enhanced Camera-Controlled Video Synt

**Questions:**

1. Is the proposed method a better way to enhance the physical world understanding ability of LLM compared with Google's SayCan for example?
2. What is the main differnece between the propsed idea with LeCun's Navigation World Model?

---

### Official Review · Reviewer_muAU · 2025-11-01

**Soundness:** 1
**Presentation:** 1
**Contribution:** 2
**Rating:** 2
**Confidence:** 4

**Summary:**

EgoPlan aims to combine a one-step planner (an LMM agent) with a world model (a diffusion model), to enable embodied decision-making in virtual environments. The manuscript claims to perform both high-level task decomposition as well as low-level action output, end-to-end. The manuscript claims to generalize across ‘varying dynamics within fixed household scenarios’.

**Strengths:**

- The manuscript pursues the combination of several concepts into a single method.
- The manuscript considers compelling tasks in Embodied AI. This is appealing as it draws attention to the limitations of then-powerful VLMs in behavior grounding in the real world.
- The manuscript provides detailed prompts for the LMM.

**Weaknesses:**

- L23-25: The manuscript states, “By using LMM, we can output text actions, using a series of mechanisms such as reflection to perform high-level task decomposition and low-level action output end-to-end.”. In L181-182, the manuscript states, “In addition, the agent uses encapsulated skills as actions, such as moving forward, turning, and grabbing objects.”. How is this realized, when the method relies on encapsulated primitives, such as move forward, turn left/right, and snap-grab something? The method does not seem to capture low-level actions at all?
- Several places in the manuscript lack clarity and should be re-written: L48-57, L59-62, L77-80, L320-323.
- L54-57: The manuscript needs to be precise about what it means by ‘varying dynamics’ and could already give hints about how a principled evaluation, given variation in the dynamics, is achieved.
- L59-62: The writing could be clearer here. The paragraph seems to assume that the reader is already familiar with the architecture of the system, whereas the overall structure has not been introduced yet. The manuscript should give a brief overview of all the main components of the system, with sufficient rationale for including each one (based on concrete limitations highlighted about the prior art.
- L74: “trajectorys” —> “trajectories”
- L77-80: The writing could be made much clearer here; it is hard to glean what EgoPlan’s main contribution is or the rationale behind each decision. Currently, the paragraph risks just deluging the reader with a bunch of different concepts (dynamics models, optical flow, LoRA, few-shot generalization) without making a strong cases for them.
- Section 2: The manuscript states, “In this section, we present a brief overview of related work. More discussions are in Appendix B.” I would strongly discourage the dismissal of a detailed literature review; a good literature review positions the manuscript effectively, highlights the limitations of the related work that the present manuscript seeks to address, and justifies the strong baselines that the manuscript will use in its experiments. Without any of that, the manuscript is left without sufficient positioning, lacks justification for its experiments, has reduced support for its claims.
- L125-128: This limitation seems disconnected from the claims, experiments, and methodology in the paper.
- Section 3: Missing comparisons with other datasets. If the manuscript wishes to claim a dataset as a contribution, further justification should be provided for why the community should use it. Conventionally, this justification is provided by comparing the properties of the proposed dataset with those of other datasets.
- L296-297: The manuscript states, “We plan to investigate the impact of different subgoal types on task performance (Section 5.4).” This analysis seems to be missing?
- Section 5.2: More description is needed in the main content of the paper; the reader will not be able to obtain a reasonable understanding of the tasks, without digging through the appendix. This lack of clarity detracts from the flow of the paper.
- Table 3: The comparisons provided in the paper are quite old and thus do not  provide meaningful insights. Since 2023, several works have leveraged similar strategies with more powerful models on the same or similar benchmarks highlighted by this manuscript. At the very least, I would recommend more meaningful comparisons before I could recommend this paper for acceptance. Some more recent works in tabletop manipulation (https://arxiv.org/pdf/2306.15724), navigation (https://arxiv.org/pdf/2412.14480), and mobile manipulation (https://arxiv.org/pdf/2305.05658).
- L467-469: Quantitative results seem to be missing.

**Questions:**

I have no additional queries. I would like to receive individual responses to the above points on the limitations of the manuscript.

---

### Official Review · Reviewer_bxYz · 2025-11-02

**Soundness:** 2
**Presentation:** 2
**Contribution:** 2
**Rating:** 2
**Confidence:** 4

**Summary:**

The paper considers embodied agents using large multi-modal models (LMMs) for planning long horizon tasks (in varying household scenarios).  The approach is called EgoPlan, emphasizing ego-centric planning.  The approach incorporates several ideas and includes an image generator with optical flow prediction. The authors develop a new dataset for navigation built from the VirtualHome sim, and carry out experiments.

**Strengths:**

The paper addresses an important problem, and develops a learning architecture, which is used in a simulated home environment.  The objective is long horizon based on text to action.

The work brings in an optical flow detection idea to better predict visual images based on a generator.  Optical flow has other applications and so this work could have other applications.

There are many experiments and ideas presented. The authors have collected a dataset in the simulation environment that includes motion frames.

The paper addresses transfer learning into a similar but different indoor environment.

**Weaknesses:**

The paper is very difficult to follow and has several ideas and drawing from many methods, and the combination is not clearly explained or smoothly presented.  The variety of methods and experiments are interesting but lack focus. Core ideas are pushed to appendices, which are disjoint and overused and detract from the overall flow and conveying ideas.  The paper is not self-consistent:  For example, in section 4, the optical flow generator is good for the next frame, but not good for predicting subgoal observations.  Later, in Section 5.3 the optical flow cannot be obtained from current to next tilmestep.  So the time scale for optical flow prediction and overall use is very confusing for the reader.

The transfer learning to different simulation environments is interesting. However, the optical flow generator is very highly dependent on training in the initial environment, and generalization here means that the new environment is very visually similar. Robustness to variations in motion velocity, lighting, and other variables isn’t clear, and the paper makes simplifying assumptions about these. This could lead to a significant sim-to-real gap, for example.

The authors suggest that only 2 potential solutions to the planning problem exist (introduction paragraph 2), fine-tuning on state-action, or employing pre-trained world models.  However, this seems too narrow and  missing work such as in the robotic control community on validation of plans and replanning, prompt modification, online learning and map updating, and object understanding semantically linked with location types.

It isn’t clear that there is any physical manipulation here, instead assuming that a set of grasping, etc., is built-in on command.

It’s stated that the generation model is fed to the LLM multiple times for choosing among these for the best. The complexity, choices, and tradeoffs here don’t seem clear.

**Questions:**

Optical flow has a long history in robotics for perception.  What connections are there?

It would be useful to characterize the overall computational complexity.

---

### Official Review · Reviewer_A5t9 · 2025-11-04

**Soundness:** 1
**Presentation:** 1
**Contribution:** 2
**Rating:** 2
**Confidence:** 2

**Summary:**

The paper proposes EgoPlan, an embodied agent pipeline that: (i) trains a conditional diffusion “world model” to predict the next egocentric frame (x_{t+1}) from the current frame (x_t) and a text action (a_t); (ii) augments the diffusion model with optical-flow conditioning (flow predicted via a VQ-GAN; injected with ControlNet); (iii) adapts visual style to new simulators via LoRA; and (iv) uses an LMM (e.g., GPT-4V) as a one-step planner that ranks candidate actions by comparing diffusion rollouts against a (text or image) subgoal and picking the action “closest” to the goal. The paper also introduces VH-1.5M, an egocentric dataset generated in VirtualHome. Experiments report FID/user-study for next-frame prediction, optical-flow AEE, “average completed subtasks” on 12 VirtualHome tasks, and an HM3D ObjectNav result in Habitat 2.0.

**Strengths:**

- Ambitious system integration across LMMs, diffusion, ControlNet, and optical flow.

- A new egocentric dataset (VH-1.5M) could have reuse value if released with splits and documentation.

- Clear high-level motivation to improve physical grounding of LMM planning via visual lookahead.

**Weaknesses:**

- Mathematical errors in core equations (diffusion loss uses wrong variable/schedule; planning objective malformed and metric undefined).

- Methodologically incremental relative to prior “world-model + planner” lines (e.g., SuSIE/VLP/RoboDreamer); one-step lookahead limits scope.

- Experimental design gaps. Primary metric is “average completed subtasks” rather than standard SR/SPL; success rates are low and relegated to the appendix; runtime/throughput absent despite heavy per-step compute (diffusion rollouts for all actions + LMM scoring).


- Reproducibility concerns. Heavy reliance on closed LMMs and prompts without ablations or stability analysis; dataset/code/weights not available at review time (or only promised).

**Questions:**

- Clarify POMDP claims: is there any belief/state estimation; any information-gathering objective?
- The equations conflate environment time (t) with diffusion step (k), and use unclear notation. Can you carefully write the math correctly?

---

### Note · Authors · 2025-11-22

I have read and agree with the venue's withdrawal policy on behalf of myself and my co-authors.